# Nuclei determine the spatial origin of mitotic waves

**Felix E Nolet[1†], Alexandra Vandervelde[1†], Arno Vanderbeke[1,2†], Liliana Piñeros[1†], Jeremy B Chang[3], Lendert Gelens[1]***

[1]Laboratory of Dynamics in Biological Systems, Department of Cellular and Molecular Medicine, Faculty of Medicine, KU Leuven, Leuven, Belgium; [2]MeBioS - Biosensors Group, Department of Biosystems, KU Leuven, Leuven, Belgium; [3]Department of Pharmaceutical Chemistry, University of California, San Francisco, United States

**Abstract** Traveling waves play an essential role in coordinating mitosis over large distances, but what determines the spatial origin of mitotic waves remains unclear. Here, we show that such waves initiate at pacemakers, regions that oscillate faster than their surroundings. In cell-free extracts of *Xenopus laevis* eggs, we find that nuclei define such pacemakers by concentrating cell cycle regulators. In computational models of diffusively coupled oscillators that account for nuclear import, nuclear positioning determines the pacemaker location. Furthermore, we find that the spatial dimensions of the oscillatory medium change the nuclear positioning and strongly influence whether a pacemaker is more likely to be at a boundary or an internal region. Finally, we confirm experimentally that increasing the system width increases the proportion of pacemakers at the boundary. Our work provides insight into how nuclei and spatial system dimensions can control local concentrations of regulators and influence the emergent behavior of mitotic waves.

**\*For correspondence:**
lendert.gelens@kuleuven.be

[†]These authors contributed equally to this work

**Competing interests:** The authors declare that no competing interests exist.

## Introduction

Traveling waves are often used in nature to transmit information quickly and reliably over large distances (*Cross and Hohenberg, 1993*; *Tyson and Keener, 1988*; *Gelens et al., 2014*; *Beta and Kruse, 2017*; *Deneke and Di Talia, 2018*). For example, action potentials are well known to propagate along the axon of a neuron (*Hodgkin and Huxley, 1952*), but a wealth of other biological processes have been shown to be coordinated via traveling waves (*Winfree, 1987*; *Dawson et al., 1999*; *Loose et al., 2008*; *Chang and Ferrell, 2013*; *Deneke et al., 2016*; *Prindle et al., 2015*; *Bement et al., 2015*; *Fukujin et al., 2016*). In particular, cell cycle oscillations also self-organize via mitotic waves in a spatially extended system (*Chang and Ferrell, 2013*; *Deneke et al., 2016*). Such waves that coordinate cell division in space are especially relevant in the large developing eggs (ranging from ≈ 100 μm to ≈ 1 mm in diameter) that are laid externally by insects, amphibians, and fish, because they are too large to be synchronized by diffusion alone (see *Box 1*). While several studies have addressed the potential biochemical mechanisms of mitotic waves (*Chang and Ferrell, 2013*; *Deneke et al., 2016*; *Vergassola et al., 2018*), what determines the spatial origin of mitotic waves remains unclear.

Here, we address this open question using cell-free extracts made from eggs of the frog *Xenopus laevis*, which exhibit biochemical cell cycle oscillations *in vitro* that are similar to those found *in vivo* (*Murray, 1991*). We find that mitotic waves originate at nuclei, which act as so-called pacemakers, regions that oscillate faster than their surroundings (*Kuramoto, 1984*). While previous studies have suggested centrosomes or nuclei to serve as pacemakers (*Chang and Ferrell, 2013*; *Ishihara et al., 2014*), their role in organizing mitotic waves has not been empirically demonstrated. We provide evidence that nuclei serve as pacemakers, both in the absence and presence of centrosomes. Having

the nucleus setting the pace of the cell cycle may help ensure proper DNA replication prior to initiation of mitosis. If the pacemaker were elsewhere, the decision to divide might be decoupled from DNA replication, leading to division occurring before DNA replication completes. We postulate that nuclei can concentrate cell cycle regulators, leading to faster cell cycle oscillations at those nuclear locations. Nuclei and their spatial positioning, which is affected by the spatial dimensions of the system, determine how the cell cycle is coordinated in space and time.

By monitoring mitotic waves in Teflon tubes using time-lapse microscopy (see *Box 2*), we find that pacemakers are often located near nuclei that are brighter due to increased import of exogenously added GFP-NLS. We show that the generation of such pacemakers does not require centrosomes and explore the influence of nuclear density and nuclear import strength on cell cycle period and pacemaker wave formation. Based on these observations, we then develop a theoretical model where nuclei play an active role in concentrating cell cycle regulators. This concentration decreases the period of oscillation around the nuclei. Our modeling shows that the distribution of regulators depends on the nuclear positioning and spatial dimensions of the system, with thicker tubes having a larger tendency to concentrate cell cycle regulators at the boundaries (i.e. outer edges of the tube). Using both numerical simulations and experiments, we go on to show that mitotic waves can originate from the system interior or from the system boundary, depending on the spatial dimensions of the system. These observed dynamics are the result of competition between waves originating from different pacemaker regions, where the relative strength of the pacemakers in the interior and at the boundary is determined by the system dimensions.

## Results

### Nuclei serve as pacemakers to organize mitotic waves

We reconstituted mitotic waves *in vitro* according to Chang and Ferrell (*Chang and Ferrell, 2013*; *Chang and Ferrell, 2018*). We loaded cycling extracts in a 100 μm wide Teflon tube and used green fluorescent protein with a nuclear localization signal (GFP-NLS) to image mitotic waves (see *Box 2*). This approach allows visualization of regular oscillations between interphase and mitotic phase. In interphase, nuclei form spontaneously in the extract supplemented with sperm chromatin. These nuclei then import GFP-NLS. In mitosis, the nuclear envelope breaks down and GFP is no longer localized to nuclei. Mitotic waves can be observed by the disappearance of nuclei in a wave-like fashion. Waves become apparent after a couple of cell cycles and they self-organize so that they emerge from more clearly defined foci (see *Figure 1A*, *Figure 1—video 1*). The origin of the wave (point P) was determined as the intersection of straight lines drawn through the points where the nuclei disappear (see orange curve and *Figure 1—figure supplement 1*). The wave at cell cycle 5–6 was found to propagate with a speed of ∼ 20 μm/min.

We noticed that the mitotic wave originated close to a nucleus that is considerably brighter than the surrounding nuclei (*Figure 1A*). We hypothesized that a region with higher GFP-NLS intensity correlates with a higher local oscillation frequency, serving as a pacemaker that organizes the mitotic wave. We therefore analyzed the spatial GFP-NLS intensity profile, the spatial profile of cell cycle periods, and the internuclear distance (*Figure 1B*). As a brighter nucleus has taken up more GFP-NLS, we reasoned that it similarly concentrates cell cycle regulators that lead to a local increase in the cell cycle frequency. We directly correlated this with the local period, which indeed showed that this region oscillated faster (*Figure 1B*). To further understand why certain nuclei were brighter, we explored whether their environment had any particular characteristics. We characterized the distance between the different nuclei and found that they were typically separated by 150–200 μm (*Figure 1—figure supplement 2*). However, we found that the brightest nucleus is also most separated from its neighboring nuclei (*Figure 1B*). This finding is consistent with the idea that nuclei increase their oscillation frequency by concentrating cell cycle regulators, as they have a larger pool of regulators in their surroundings to import. We analyzed the spatial GFP-NLS intensity profile and the internuclear distance for nine other experiments where we could clearly identify nuclei and mitotic waves. Overall, in 90% of the analyzed experiments the pacemaker location was well predicted by the region with the highest GFP-NLS intensity and/or the region where nuclei were most separated from their neighboring nuclei (*Figure 1A,B*, *Figure 1—figure supplement 3*, *Figure 1—figure supplement 4*). The total nuclear GFP-NLS intensity was also found to be a better indicator of the

## Box 1. Spatial cell cycle coordination in early frog and fly embryos.

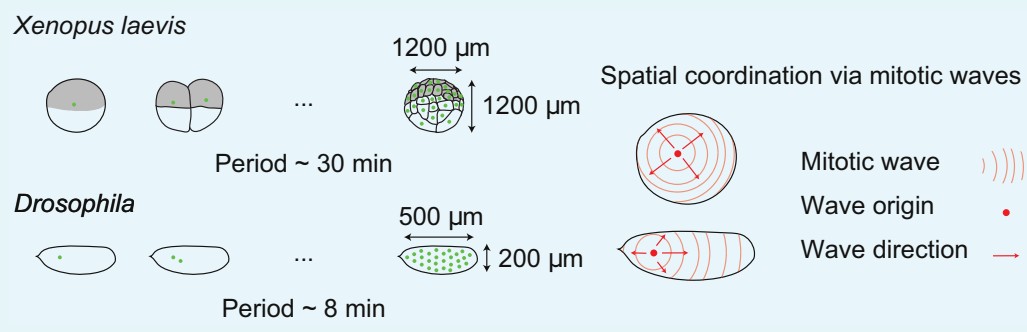

**Box 1—figure 1.** Spatial cell cycle coordination in early frog and fly embryos.

Insects, amphibians, and fish lay their eggs externally. After fertilization, all of these organisms need to go from a single, large cell to several thousands of somatic-sized cells that then further develop into an adult animal. They do so by carrying out multiple rounds of rapid cleavages following fertilization (*Foe and Alberts, 1983*; *Olivier et al., 2010*; *Farrell and O'Farrell, 2014*; *Anderson et al., 2017*). While in the *Xenopus laevis* frog embryo a new cell membrane is formed around each nucleus (green dots in figure), this is not the case in the *Drosophila* fly embryo, leading to a multinucleated syncytium. Due to the large size of these embryos, diffusion is not fast enough to spatially coordinate the cell cycle (*Gelens et al., 2014*; *Deneke and Di Talia, 2018*). There is thus a need for an alternative mechanism to coordinate cell cycle processes over large distances. Mitotic waves have been observed in the shared cytoplasm of large *Xenopus* cells (*Chang and Ferrell, 2013*) and the syncytium of the *Drosophila* embryo (*Deneke et al., 2016*; *Vergassola et al., 2018*). Such wave-like propagation of the mitotic state is believed to help spatially coordinate cell cycle progression, yet the spatial origin of this mitotic wave remains unclear.

pacemaker location than the nuclear size as indicated by Hoechst staining, or than the GFP-NLS intensity normalized to the Hoechst signal (*Figure 1—figure supplement 4*).

In order to further test the role of nuclei as pacemakers, we explored alternative markers of mitotic entry that do not rely on the nuclei themselves. We repeated the experiment with a microtubule reporter, using fluorescently labeled tubulin (HiLyte Fluor 488). *Figure 1C* and *Figure 1—video 2* show that mitotic waves are also observed using such a microtubule reporter, as well as in brightfield. With these tools in hand, we set out to test how critical system parameters such as nuclear density and nuclear import strength influence the mitotic wave dynamics.

### Nuclear density and nuclear import strength control cell cycle period and mitotic wave speed

We repeated the experiment in tubes of 100 and 200 μm width for two different concentrations of added demembranated sperm nuclei (approx. 60 and 250 nuclei/μl) (*Figure 2A,B*). We found that extracts with less added sperm nuclei had a faster cell cycle (*Figure 2B*). Mitotic waves were similarly observed, but the wave speeds were initially faster than in tubes with a higher nuclear density (*Figure 2A*). The waves then slowed down to similar speeds as in the case with the higher concentration of sperm nuclei. For both nuclear densities we also found that the average cell cycle period increases over time (*Figure 2B*). Such a correlation of mitotic wave speed with cell cycle duration is consistent with a transition from sweep waves to trigger waves as the cell cycle slows down (*Vergassola et al., 2018*). An increase in cell cycle period has been linked to a decrease in ATP supply over time (*Guan et al., 2018*). An additional explanation could be that an increase in cell cycle period is related to increasing levels of DNA as it is replicated (*Dasso and Newport, 1990*). This would also explain the decreasing period when reducing the concentration of added sperm nuclei.

## Box 2. Reconstituting cell cycle oscillations using cell-free extracts.

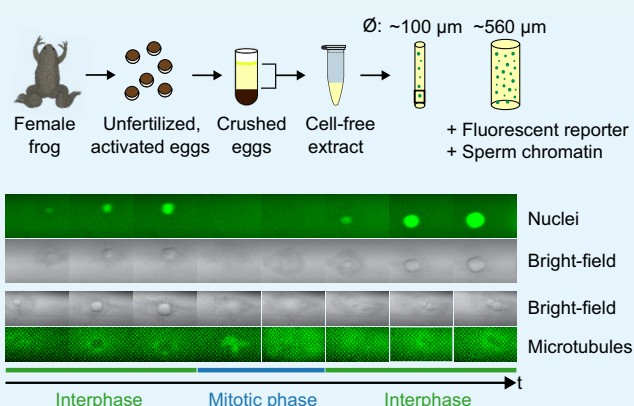

**Box 2—figure 1.** Reconstituting cell cycle oscillations using cell-free extracts.

Cell-free cycling extracts can be made from thousands of unfertilized *Xenopus laevis* frog eggs, following the protocol by ***Murray, 1991***. Cycling extracts can be supplemented with green fluorescent protein with a nuclear localization signal (GFP-NLS) and demembranated sperm nuclei. We load them in Teflon tubes of varying diameters, and image them with a confocal microscope. Under these conditions biochemical oscillations persist and drive the spontaneous formation of nuclei in the extract. Such cell cycle oscillations can be observed by the fluorescent nuclei (importing GFP-NLS) that periodically appear (interphase) and disappear (mitotic phase). Similar oscillations can be observed in bright-field and/or by using fluorescently labeled microtubules (HiLyte Fluor 488).

Interestingly, a decrease in nuclear density did not lead to a big change in the internuclear distance (*Figure 1—figure supplement 2I*). Instead, it created more and larger regions where nuclei were absent (*Figure 1—figure supplement 3*), and pacemakers were predominantly found close to these regions (*Figure 1—figure supplement 3*). Cheng and Ferrell observed a similar transition from a regular pattern of equidistantly spaced nuclei to a system with holes in *Xenopus* interphase egg extracts when decreasing the concentration of added sperm nuclei (*Cheng and Ferrell, 2019*). Next, we further decreased the nuclear density (approx. 30 nuclei/µl), such that only few nuclei remained in an entire tube. Here, we used the fluorescent microtubule reporter to visualize the spatial coordination of mitotic entry, while bright-field images were used to track the location of nuclei (*Video 1*). Mitotic waves were found to originate at the few nuclei present in the tube, and they traveled through the whole tube (several mm) at a speed of approx. 60 µm/min (*Video 1*, *Figure 2C*). In the absence of any nuclei in the tube (no added demembranated sperm nuclei), we still observed cell cycle oscillations with periods similar to extracts with low concentrations of demembranated sperm nuclei (*Figure 2—figure supplement 1*). However, no mitotic waves were observed (*Video 1*). These experiments underscore the critical role that nuclei play in changing the cell cycle period and organizing mitotic waves.

Centrosomes have also been suggested to serve as pacemakers (*Chang and Ferrell, 2013*; *Ishihara et al., 2014*), potentially by concentrating pro-mitotic factors such as Cdc25 and cyclin B (*Bonnet et al., 2008*; *Jackman et al., 2003*). Demembranated sperm nuclei are known to have associated centrioles, which give rise to centrosomes that can generate microtubule asters. In order to test whether such centrosomes are critical to generate pacemakers, we added purified DNA to the extracts, which assembled into nuclei (*Newmeyer et al., 1986*). Mitotic waves were still observed

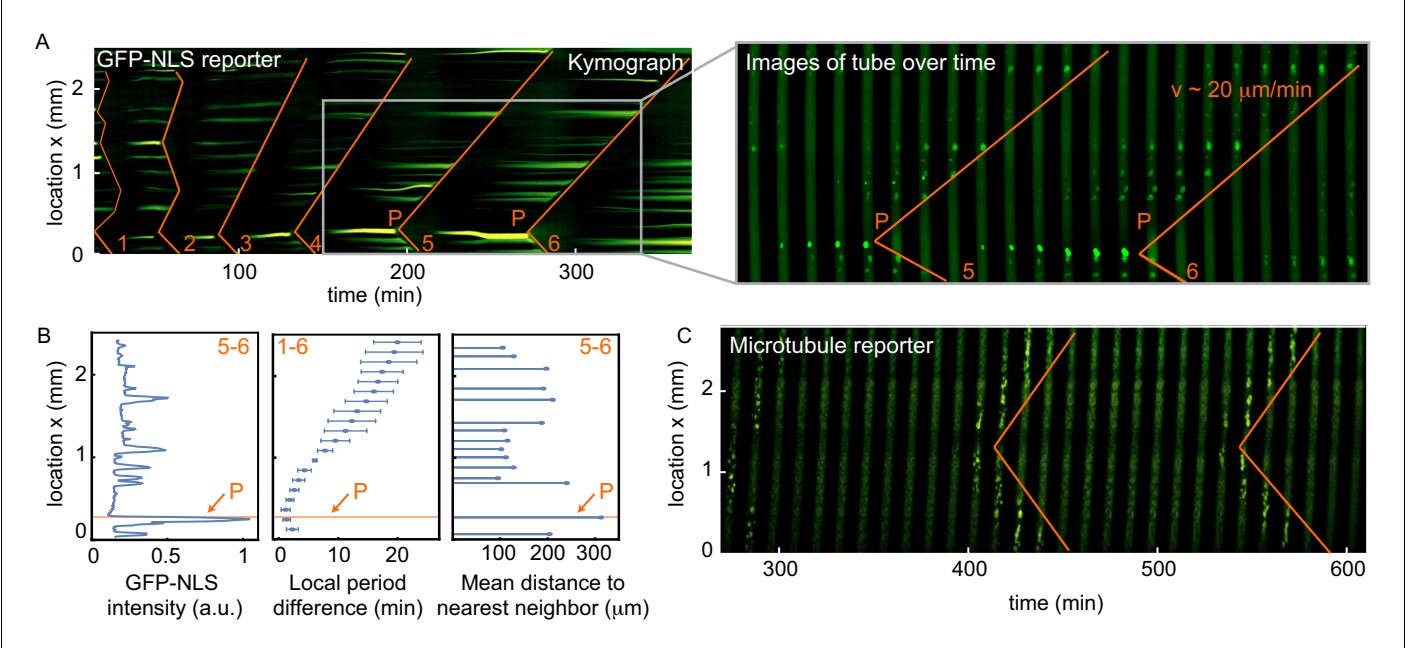

**Figure 1.** Nuclei serve as pacemakers to organize mitotic waves. (**A**) Mitotic waves (orange) in a kymograph of cell-free extract experiment in a 100 µm Teflon tube. Wave dynamics are shown for cell cycle 1–6. For each time point we reduced the data from two to one spatial dimension by plotting the maximal GFP-NLS intensity along the transverse section of the tube. In the zoom, indicated by the gray box, we show snapshots of the whole 100 µm wide tube for different time points. The pacemaker location in cell cycle six is indicated by P. Approx. 250 nuclei/µl are added. (**B**) Analysis for the experiment in A. Left: GFP-NLS intensity profile, averaged over the times between the mitotic waves in cell cycle 5 and 6. The GFP-NLS intensity is highest close to the pacemaker region P. Middle: Difference in cell cycle period (with respect to the fastest period) at different locations along the tube, averaged over cell cycle 1–6, showing that the pacemaker region oscillates fastest. Right: Mean distance from the center of each nucleus to its two nearest neighboring nuclei. The nucleus close to the pacemaker region P is most separated from its neighbors. (**C**) Mitotic waves in a 200 µm Teflon tube shown by a fluorescent microtubule reporter (HiLyte Fluor 488).

The online version of this article includes the following video and figure supplement(s) for figure 1:

**Figure supplement 1.** Methodology of image analysis of the experiments.

**Figure supplement 2.** Analysis of the experiment in panel A, quantifying the time evolution of the number of nuclei, the nuclear size, the internuclear distance, the oscillation period, the intensity of the nuclei, and the observed wave speed.

**Figure supplement 3.** Analysis of the spatial GFP-NLS intensity profile and the internuclear distances for multiple experiments.

**Figure supplement 4.** Analysis of the spatial GFP-NLS and Hoechst intensity profile and the internuclear distances.

**Figure 1—video 1.** Video of the cell-free extract experiment in panel A, B.

https://elifesciences.org/articles/52868#fig1video1

**Figure 1—video 2.** Video of the cell-free extract experiment in panel C.

https://elifesciences.org/articles/52868#fig1video2

---

indicating that DNA alone is sufficient to create pacemaker-generated mitotic waves without a need for centrosomes (*Figure 2D*, *Figure 2—video 1*).

As we hypothesize that the import of cell cycle regulators into the nucleus locally changes the cell cycle period, we decided to manipulate the nuclear import strength. We used the nuclear import inhibitor importazole, which is an inhibitor of importin-$\beta$ transport receptors. Increasing levels of importazole were found to increase the cell cycle period and slowed down the formation of nuclei (*Figure 2E,F*). Mitotic waves were still observed with similar speeds for lower concentrations of importazole, while concentrations higher than 60 µM abolished the formation of nuclei and mitotic waves. Increasing inhibition of nuclear import was also found to lead to smaller nuclei with dimmer levels of GFP-NLS (*Figure 2G*). When nuclei became very small (i.e. for 40 µM importazole), it took long for the extract to start cycling and mitotic waves were lost (*Figure 2F*, *Figure 2—video 2*). We also indirectly manipulated nuclear formation by inhibiting the kinesin Eg5 using S-Trityl-L-cysteine (STLC), which interferes with the proper formation of microtubule structures. We found that increasing concentrations of STLC gradually increased the average cell cycle period (*Figure 2—figure*

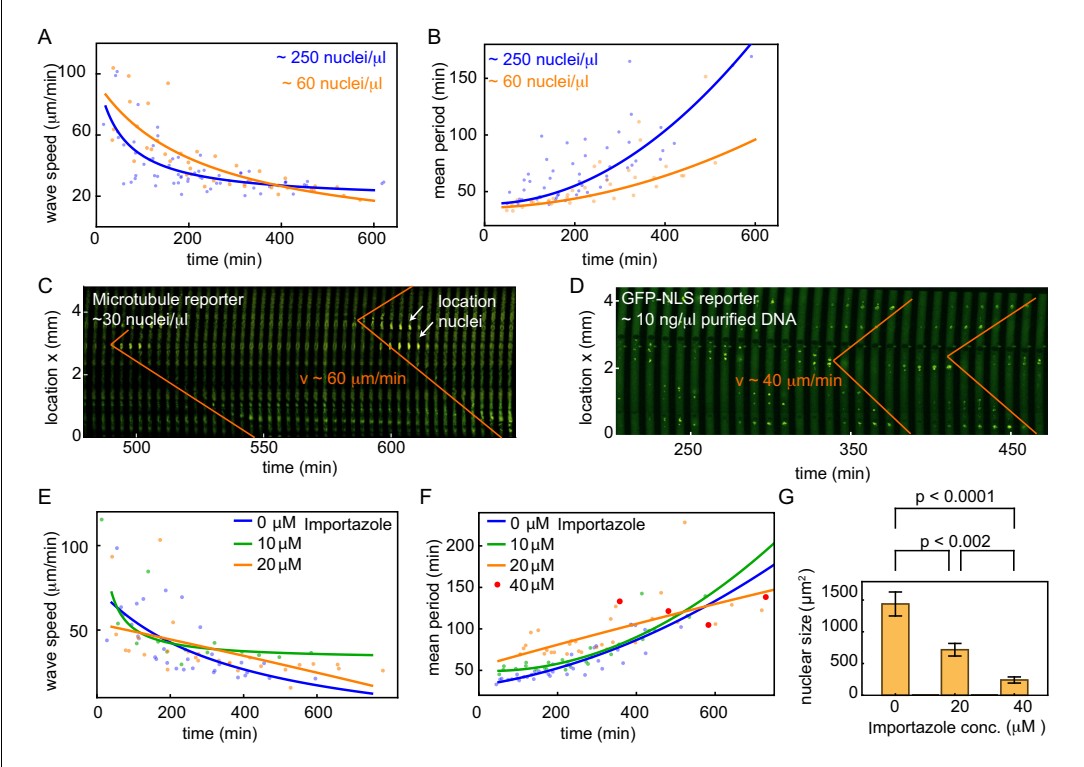

**Figure 2.** Nuclear density and nuclear import strength control cell cycle period and mitotic wave speed. (**A,B**) Wave speed (**A**) and cell cycle period (**B**) over time obtained for N = 19 analyzed 100 and 200 μm Teflon tube experiments using the GFP-NLS reporter. Results are pooled from 11 different cell-free extracts for two different nuclear concentrations: ≈ 60, and ≈ 250 nuclei/μl. Each plotted point corresponds to the minimal wave speed or average cell cycle period in a single cell cycle of a single tube experiment. (**C**) Mitotic waves in a 200 μm Teflon tube using a GFP-MT reporter with few nuclei (≈ 30 nuclei/μl). Nuclear locations are identified in bright-field and indicated here. (**D**) Mitotic waves in a 200 μm Teflon tube using a GFP-NLS reporter with ≈ 10 ng/μl of added purified DNA. (**E,F**) Wave speed (**E**) and cell cycle period (**F**) over time obtained for N = 16 analyzed 200 μm Teflon tube experiments using the GFP-NLS reporter. Results are pooled from two different cell-free extracts for four different concentrations of the nuclear import inhibitor importazole: 0, 10, 20, 40 μM. Nuclear concentration: ≈ 250 nuclei/μl. Each plotted point corresponds to the minimal wave speed or average cell cycle period in a single cell cycle of a single tube experiment. (**G**) Mean nuclear size in the presence of varying concentrations of the nuclear import inhibitor importazole: 0, 20, 40 μM. Two tube experiments were analyzed per condition, which gave us nuclear sizes for 75, 62, and 25 nuclei, for 0, 20, 40 μM importazole, respectively. Error bars are one standard deviation of the mean.

The online version of this article includes the following video and figure supplement(s) for figure 2:

**Figure supplement 1.** Influence of nuclear density on cell cycle period.

**Figure supplement 2.** Influence of Eg5 kinesin inhibitor on wave speed and cell cycle period.

**Figure 2—video 1.** Video of the cell-free extract experiment in panel D.

https://elifesciences.org/articles/52868#fig2video1

**Figure 2—video 2.** Video of the cell-free extract experiment in panels E-G.

https://elifesciences.org/articles/52868#fig2video2

supplement 2). Here too, nuclei no longer formed and mitotic waves were no longer observed when STLC was present in too high concentrations (approx. 40 μM STLC). Overall, these findings confirm that nuclear import processes are important in organizing mitotic waves. They ensure that nuclei are able to introduce sufficient spatial heterogeneity in cell cycle period to generate clear mitotic waves.

# A computational model where nuclei spatially redistribute cell cycle regulators predicts the location of pacemaker regions

Based on our experimental observation showing that brighter nuclei serve as pacemakers, we set out to develop a theoretical model that describes how GFP-NLS and other proteins can be spatially redistributed by nuclei. A sketch illustrating such a model is shown in *Figure 3A*. The system toggles between interphase and mitosis with a fixed period. During interphase, nuclei form and nuclear proteins (such as GFP-NLS) are actively imported into the nucleus. During mitosis, the nuclear envelope breaks down and proteins are free to diffuse away. We implemented the competing import and diffusion processes using a generic partial differential equation (PDE) model that describes the evolution of the concentration $C$ of nuclear protein, such as GFP-NLS (for details on this model, see Appendix 1). These competing processes are relevant for all proteins that localize to the nucleus. For example, it is known that APC/C is mostly localized in the nucleus, and Wee1 and Cdc25 are actively transported between cytoplasm and nucleus during the cell cycle (*Baldin and Ducommun, 1995*; *Arnold et al., 2015*). Such relocalization of cell cycle regulators can locally change the cell cycle oscillation frequency. Note that different different proteins can have opposing effects. For example, while increasing activity of Wee1 and APC/C tend to increase the cell cycle oscillation period, increasing Cdc25 activity leads to faster oscillations (*Novak and Tyson, 1993*; *Tsai et al., 2014*). Our experiments thus suggest that the overall effect of increasing nuclear import is to decrease the cell cycle period.

We start by studying the simplest case of a single nucleus in the center of a one-dimensional domain (see *Figure 3B*). We defined the spatial range of attraction around the nucleus to be approx. 100 μm, such that it is consistent with the so-called *nuclear domain*, a subdomain of the cytoplasm in which spatial constraints show an effect on nuclear growth (*Hara and Merten, 2015*). In *Xenopus* cell-free extract, this nuclear domain has a diameter of approx. 170 μm (*Hara and Merten, 2015*). The term nuclear domain was originally introduced to describe the surroundings of evenly spaced nuclei in syncytial muscle fibers and *Drosophila* embryos (*Landing et al., 1974*; *Telley et al., 2012*). In *Drosophila* embryos, the nuclear domain (also called energid) is approx. 30 μm for nuclei which are approx. 5–10 μm in diameter (*Chen et al., 2012*; *Telley et al., 2012*). In our experiments we find an internuclear distance of approx. 150 μm for nuclei of approx. 40 μm in diameter (*Figure 1—figure supplement 2G-I*). *Figure 3B* shows that proteins quickly build up in the nuclear region in the early phase of the import period and then the proteins quickly disperse after nuclear envelope breakdown. As expected, when averaging the concentration profile over one cell cycle, we find that the time-averaged concentration $C_{avg}$ peaks around the nucleus (see red area in *Figure 3B*), defining a pacemaker at the nucleus.

While a typical cell contains a single nucleus, the cell-free extract experiment shown in *Figure 1* consists of many distributed nuclei. From the experimental data, we calculated how far different nuclei are separated from each other, finding

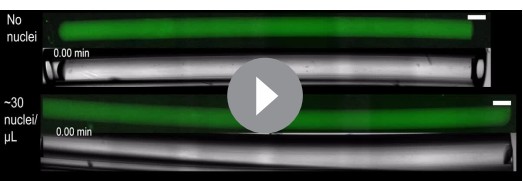

**Video 1.** Video of cell-free extract experiment in a 200 μm wide Teflon tube imaged in bright-field and using a fluorescent microtubule reporter (HiLyte Fluor 488). The experiment on the bottom (see also *Figure 1C*) has few nuclei (≈ 30 nuclei/μl), while no nuclei are added in the experiment on the top. In the presence of few nuclei, mitotic waves originate from those nuclei and propagate through the whole tube. In the absence of nuclei, no mitotic waves are observed to travel through the tube. Scale bar is 200 μm.
https://elifesciences.org/articles/52868#video1

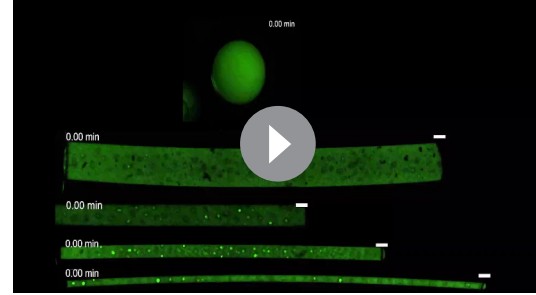

**Video 2.** Video of cell-free extract experiments in Teflon tubes of varying diameters (≈ 100, 200, 300 and 560 μm wide) and a thin droplet of ≈ 1 mm wide. Imaging is done with the GFP-NLS reporter. Mitotic waves are found to originate from the boundary as the system becomes wider. Scale bar is 200 μm.
https://elifesciences.org/articles/52868#video2

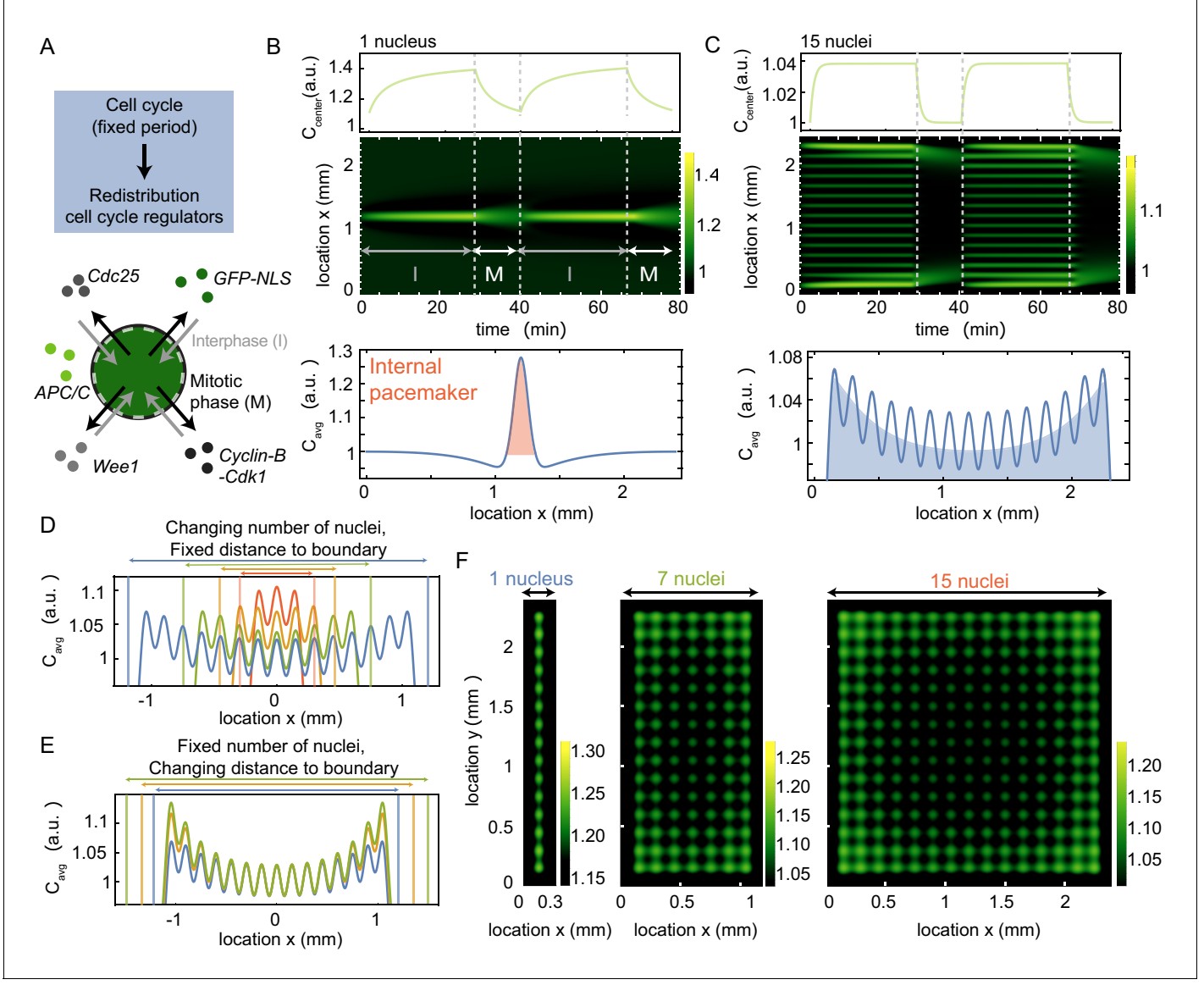

**Figure 3.** A model where nuclei spatially redistribute cell cycle regulators predicts the location of pacemaker regions. (A) Schematic of the two phases of the model, interphase (import of regulators) and mitotic phase (diffusion). The cell cycle has a fixed period, which controls the periodic spatial redistribution of regulators. (B) Time evolution of *Equation (8)* in Appendix 1 in one spatial dimension for one nucleus, with the concentration C at the center of the domain shown in the top panel. The profile below is the time average of the intensity over one cell cycle period ($C_{avg}$), where the red area highlights the build-up of cell cycle regulators close to the nucleus. The intensity $C$ is normalized such that $\frac{1}{L}\int_{x=0}^{x=L} C = 1$. Parameters: $\epsilon = 4 \cdot 10^4$ µm$^3$/min, $\sigma = 60$ µm, $\alpha = 0.7$, $T = 40$ min, $D = 600$ µm$^2$/min and constant initial condition $C = 1$. Domain size $L$ is 2400 µm. (C) Same as B, but now for 15 nuclei, where the time-averaged profile $C_{avg}$ shows an overall build-up of regulators towards the boundary (see blue shaded area). (D) Same as C, but now varying the number of nuclei in the system, while keeping the distance of the outer nucleus to the system boundary constant. The total system size changes as a result of the changing number of nuclei. (E) Same as C, but now varying distances of the outer nucleus to the system boundary ($d_b$), while keeping the number of nuclei constant. The total system size changes as a result of the changing distance to the boundary $d_b$. (F) Same as B and C, but in a rectangular system of two spatial dimensions. The length of the system is fixed to 2400 µm, while the width of the system increases from 300 µm (with one nucleus) to 2400 µm (with 15 nuclei). The time-averaged profile $C_{avg}$ is plotted, again illustrating the overall build-up of regulators towards the boundary.

The online version of this article includes the following figure supplement(s) for figure 3:

**Figure supplement 1.** Influence of the distance of the outer nuclei to the system boundary on the build-up of regulators at the boundary.

**Figure supplement 2.** Influence of system parameters on the build-up of regulators at the boundary.

**Figure supplement 3.** Influence of deviations to a perfect nuclear pattern in 1D on the build-up of regulators at the boundary.

**Figure supplement 4.** Influence of internuclear distance on the build-up of regulators at the boundary.

**Figure supplement 5.** Internuclear distance in tubes of varying width.

*Figure 3 continued on next page*

*Figure 3 continued*

**Figure supplement 6.** Influence of varying system widths in 2D on the build-up of regulators at the boundary.

that the distance between neighboring nuclei is typically around 150 μm (*Figure 1—figure supplement 2G–I*). Note that this distance is consistent with the typical size of a nuclear domain in *Xenopus* cell-free extract as mentioned before. Moreover, the internuclear distance is also consistent with the size of the recently characterized cell-like compartments that self-organize from homogenized interphase egg cytoplasm (*Cheng and Ferrell, 2019*). Using this information, we carried out simulations where many nuclei are equidistantly distributed over the whole domain. Such a simulation with 15 nuclei in a domain of 2.4 mm is shown in *Figure 3C*. Similarly as in the case of a single nucleus, the concentration $C$ increases during interphase at each nuclear location, while it quickly decreases during mitosis. However, nuclei close to the boundary are found to have a higher average concentration $C_{avg}$ (see blue shaded area in *Figure 3C*), which corresponds to a stronger pacemaker region at the boundary.

The build-up of regulators at the boundary is mainly attributed to the fact that nuclei in the interior of the domain compete with neighboring nuclei to attract the available proteins, while nuclei close to the boundary only have one such 'competitor'. In *Figure 3D* we verify how the number of nuclei in the system affects the average distribution of regulators, keeping the distance between the outer nuclei and the system boundary constant. Starting from the situation with 15 nuclei in *Figure 3C* (blue), we gradually decreased the number of nuclei in the system. *Figure 3D* shows that for decreasing numbers of nuclei (nine in green, five in orange, and three in red), the build-up of regulators at the boundary gradually decreases. When only having three nuclei in the system (red), the central nucleus is found to be dominant and the boundary effect is completely lost. Apart from this competition for regulators between neighboring nuclei, the location of the boundary itself could play an important role. We quantified this boundary effect by changing the distance from the outer nuclei to the system boundary ($d_b$), while keeping the number of nuclei in the system fixed (15 nuclei). *Figure 3E* shows that initially an increase in the distance to the boundary $d_b$ leads to a larger build-up of regulators at the boundary, but this increase saturates as $d_b$ becomes larger (*Figure 3—figure supplement 1*). Although the extent to which regulators build up close to the boundary also depends on the model parameters and on the exact nuclear distribution (see *Figure 3—figure supplement 2*, *Figure 3—figure supplement 3*, *Figure 3—figure supplement 4*), it was found to be a robust phenomenon. Interestingly, however, randomly removing a few nuclei within the domain could abolish the build-up of regulators at the boundary. Instead, proteins build up close to the nuclei adjacent to the gaps (*Figure 3—figure supplement 3*).

Finally, we expanded our model to two spatial dimensions. We considered rectangular domains of varying aspect ratios, keeping one side fixed in length, while varying the other side in width. The long side was chosen the same as in *Figure 3C* in which we again define 15 nuclei. We then explored the effect of different widths with increasing rows of nuclei, see *Figure 3F*. The number of rows of nuclei was based on the experimental observation that wider systems support more nuclei and that those nuclei are separated by the same internuclear distance as in the thin tubes (*Figure 3—figure supplement 5*). Similarly as in the one-dimensional case, we observe that nuclear cell cycle regulators build up at the edges of the domain. This effect was particularly strong along the longest side of the rectangle, and strikingly, it became more pronounced as the width of the domain increased (see *Figure 3F*, *Figure 3—figure supplement 6*).

## Multiple pacemakers compete to define the direction of mitotic waves

Based on the model in the previous section, we were able to make predictions of how different nuclear patterns can lead to well-defined spatial distributions of cell cycle regulators. However, transitions between interphase (nuclear import) and mitotic phase (nuclear envelope breakdown and diffusion) occurred with a fixed period. Here, we expand the model by introducing a dependence of the cell cycle period on the local concentration of cell cycle regulators (see details in Appendix 1). In this way a spatial heterogeneity in the concentration of cell cycle regulators leads to a corresponding spatial frequency profile. In general, one expects that such spatial heterogeneities in the cell cycle period create multiple waves. These waves typically propagate into the surrounding medium and

compete with each other until the pacemaker with the highest frequency ultimately entrains the whole system (*Kuramoto, 1984*).

We used this model to explore the dynamics of a pattern of 20 equidistantly distributed nuclei in a domain of 4.2 mm. *Figure 4A,D* shows that on average cell cycle regulators build up close to the boundary, similarly as in *Figure 3C*. In the current model, however, this build-up of regulators also leads to a decreased cell cycle period at the boundary. Such a pacemaker region close to the boundary then sends out waves that gradually control the whole domain and they travel more quickly for larger diffusion strengths $D$. We then gradually increased the strength of nuclear import of the three most central nuclei, which on average led to an increased concentration of cell cycle regulators here. For moderate increases in nuclear import strength, two waves compete with one another. A boundary-driven wave and a wave coming from the interior of the domain coexist (*Figure 4B*). Further increasing the nuclear import strength, waves no longer emerged from the boundary and were entirely controlled by the central region of "bright" nuclei (*Figure 4C*).

Next, we removed a nucleus from the center of the domain. Previously, for fixed cell cycle periods, we found that removing nuclei abolished the build-up of regulators at the boundary and proteins localized close to the nuclei adjacent to the gaps (*Figure 3—figure supplement 3*). *Figure 4E*

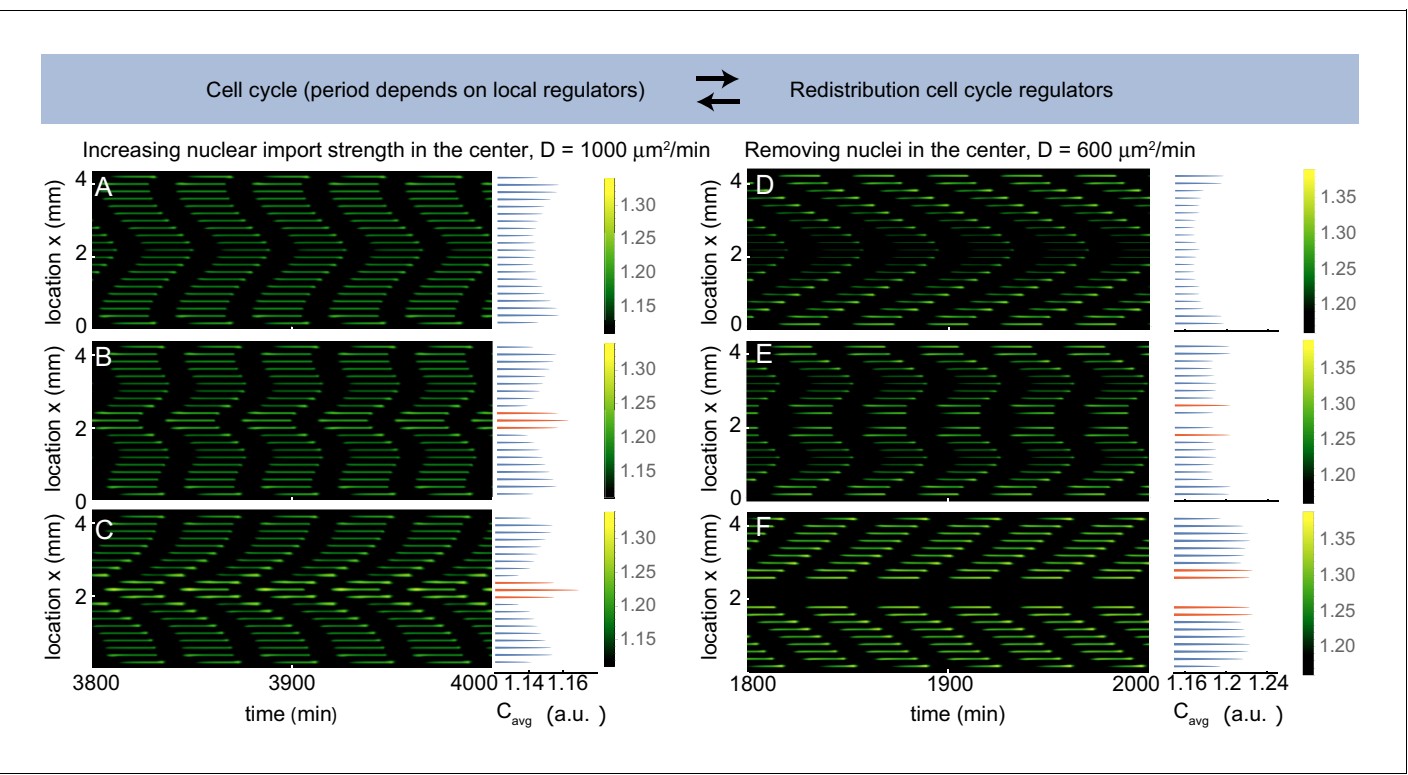

**Figure 4.** Multiple pacemakers compete to define the direction of mitotic waves. Time evolution of *Equation (21)* in Appendix 1 in one spatial dimension. The profile on the right is the time average of the intensity over one cell cycle period ($C_{avg}$). The intensity $C$ is normalized such that $\frac{1}{L}\int_{x=0}^{x=L} C = 1$. Parameters: $\epsilon = 6 \cdot 10^4$ µm$^3$/min, $\sigma = 60$ µm, $\alpha = 0.7$ and initial condition $C = 1$. (A-C) $D = 1000$ µm$^2$/min, domain size $L$ is 4400 µm including 21 nuclei separated by 200 µm. $\beta$ is defined as a factor by which the nuclear import strength $\epsilon$ is increased in the middle nucleus (see Appendix 1). $\beta$ is 1 (A), 1.04 (B) and 1.08 (C). Upon increasing the nuclear import strength of the middle nucleus, a transition is observed from boundary-driven waves (A) to waves coming from an internal pacemaker (C). The internal pacemaker region has a higher average concentration of the regulator $C$, as indicated in orange. For intermediate values of $\beta$ both types of waves coexist (B). (D-E) $D = 600$ µm$^2$/min, $\beta = 1$, domain size $L$ is 4400 µm. When 21 nuclei are regularly separated by 200 µm, a boundary-driven wave is observed (D). While removing the middle nucleus leads to the coexistence of boundary-driven waves and a waves coming from an internal pacemaker region close to the introduced gap (E), removing three of the middle nuclei abolishes the boundary-driven wave and only the wave coming from the internal pacemaker region persists.

The online version of this article includes the following figure supplement(s) for figure 4:

**Figure supplement 1.** Competing pacemakers in known PDE models for cell cycle oscillations reproduce similar mitotic wave dynamics.

**Figure supplement 2.** Boundary-driven waves can exist in spatially-extended systems based on different types of oscillators.

indeed illustrates that there is an increased concentration of regulators close to the central gap, but a build-up of regulators close to the boundary also persisted, such that two competing waves were found. We then removed two more nuclei from the center (*Figure 4F*), which caused the central pacemaker region to send out a wave that controlled the whole domain. The fact that increasing nuclear import strengths and the absence of nuclei within a nuclear pattern both lead to the creation of waves from a nearby location is consistent with the experimental observations reported in *Figure 1*, *Figure 1—figure supplement 3*).

We wondered whether these dynamics of competing pacemakers are specific to this particular computational model that includes nuclear import and diffusion processes. Therefore, we also implemented known PDE models of cell cycle oscillations (Appendix 2), where we define two pacemaker regions (see *Figure 4—figure supplement 1G–I*): an internal pacemaker and a boundary pacemaker region. We carried out simulations continuously changing the relative strength of both pacemaker regions by increasing the difference in cell cycle period. We found a gradual transition from boundary-driven dynamics to internal pacemaker-driven dynamics (*Figure 4—figure supplement 1*). Similar results were found by using the FitzHugh-Nagumo oscillator model, a general model for relaxation-type oscillatory systems (*Figure 4—figure supplement 1K*, *Figure 4—figure supplement 2*). Moreover, we found that even more sinusoidal oscillations preserved boundary-driven waves (*Figure 4—figure supplement 2*). This suggests that the generation of boundary-driven waves is largely independent of the type of oscillations, as long as the oscillation period is decreased close to the boundary.

Our findings underscore the generic character of the dynamics of multiple competing pacemakers. Pacemaker-driven traveling waves, also often referred to as target patterns, have been widely studied and they form thanks to spatial heterogeneities that locally increase the oscillation frequency. The majority of such pacemaker waves were initially observed in chemical reaction-diffusion systems where heterogeneities were introduced as dust particles that locally modified the properties of the medium (*Zaikin and Zhabotinsky, 1970*; *Zhabotinsky and Zaikin, 1973*; *Tyson and Fife, 1980*). These experimental observations triggered many other studies on both traveling waves (*Tyson and Fife, 1980*; *Kopell, 1981*; *Hagan, 1981*; *Kuramoto, 1984*; *Jakubith et al., 1990*; *Bugrim et al., 1996*; *Bub et al., 2005*; *Stich and Mikhailov, 2006*) and spiral waves (*Jakubith et al., 1990*; *Bub et al., 2002*; *Bub et al., 2005*) triggered by a pacemaker. The interaction of multiple pacemaker waves has also been analyzed (*Kuramoto, 1984*; *Walgraef et al., 1983*; *Mikhailov and Engel, 1986*; *Lee et al., 1996*; *Kheowan et al., 2007*). In general, they propagate into the surrounding medium and compete with each other until the pacemaker with the highest frequency ultimately entrains the whole system (*Kuramoto, 1984*). The existence of the transition region is therefore somewhat surprising. However, simulating the system for increasingly longer transient times, we find that the transition region where boundary-driven waves and internal pacemaker-driven waves coexist shrinks, suggesting that after infinitely long transients one pacemaker indeed controls the whole domain. Such infinite transient times are, however, less biologically relevant as the early embryonic cell cycle oscillations only persist for about 13 cycles (*Box 2*). Therefore, one would expect to observe the full range of transient pacemaker dynamics in actual biological systems.

## Wider systems lead to boundary-driven mitotic waves

Our modeling leads to several predictions. First, wider systems lead to higher concentrations of cell cycle regulators at the boundary. Such a local decrease of the cell cycle period leads to boundary-driven mitotic waves. Second, systems with intermediate width allow both internally- and boundary-driven pacemakers. Third, sparsely distributed nuclei favor internal pacemakers. Based on these three predictions, we set out to verify them experimentally.

We repeated the experiment in *Figure 1* for varying diameters of the Teflon tubes (approximately 100, 200, 300 and 560 µm) for a nuclear concentration of $\approx$ 250 nuclei/µl. A representative selection of videos corresponding to this set of experiments is shown in *Video 2* (for corresponding kymographs, see *Figure 5—figure supplement 1*). While the thinnest tube shows mitotic waves coordinated by internal pacemakers, mitotic waves are boundary-driven over the whole domain in the thickest tube. This is consistent with the first theoretical prediction that wider systems lead to boundary-driven mitotic waves. Furthermore, *Video 2* illustrates that in tubes of intermediate width (200 and 300 µm), boundary-driven waves coexist with mitotic waves that are driven by internal pacemakers. This is consistent with the second theoretical prediction. By analyzing experiments of

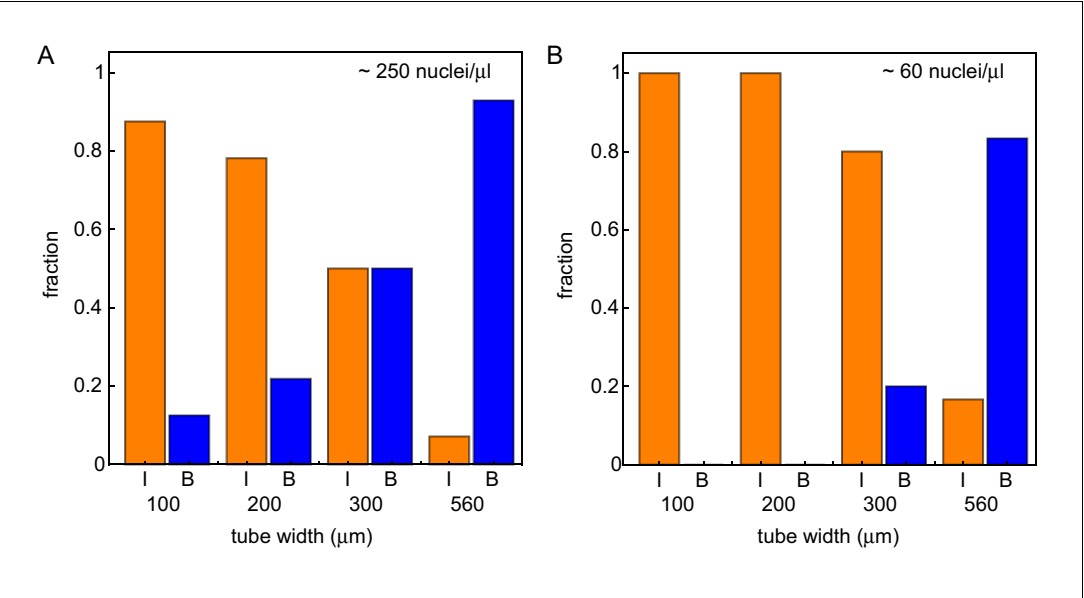

**Figure 5.** Wider systems lead to boundary-driven mitotic waves. Fraction of experiments dominated by internally-driven waves ('I') and by boundary driven waves ('B'), evaluated at the end of each of the $N = 66$ imaged tubes of varying width and varying concentration of demembranated sperm nuclei. Cases where both wave types coexist ('IB') are counted half in each category. This is done for two different concentration of demembranated sperm nuclei: ≈ 250 nuclei/μL extract (**A**) or ≈ 60 nuclei/μL extract (**B**). For panel A (**B**), results are obtained for N = 49 (17) analyzed Teflon tube experiments using the GFP-NLS reporter, and they are pooled from 23 (7) different cell-free extracts.

The online version of this article includes the following figure supplement(s) for figure 5:

**Figure supplement 1.** Kymographs of mitotic waves in tubes of varying width.

**Figure supplement 2.** Kymographs of mitotic waves in thick tubes (560 μm diameter).

**Figure supplement 3.** Analysis of all experiments, including those that did not cycle or did not show any wave dynamics.

**Figure supplement 4.** Robustness of image analysis of experiments.

**Figure supplement 5.** Wave speed and cell cycle period for varying tube width.

49 tubes of varying widths, we found these findings to be consistent (see *Figure 5A*). While the thinnest tubes have the lowest probability of finding boundary-driven mitotic waves, all of the experiments with the thickest 560 μm tubes showed boundary-driven waves (*Figure 5—figure supplement 2*). The fraction of experiments with boundary-driven wave dynamics increased smoothly with the tube width. For a more detailed analysis, see *Figure 5—figure supplement 3*.

Next, we repeated the experiments using a lower concentration of added sperm nuclei (≈ 60 nuclei/μl). This strongly decreased the probability for mitotic waves to originate from the boundary (see *Figure 5B*). We noticed that the regularity of the nuclear pattern was disrupted due to the decreased amount of nuclei. Consistent with our third theoretical prediction, the absence of neighboring nuclei was found to strengthen nearby pacemaker regions and decreased the likelihood of having pacemaker regions at the boundary (see *Figure 1—figure supplement 3*).

As boundary-driven waves were especially clear in the thickest Teflon tubes (560 μm), we wondered whether it was important for the system to be wide enough in all three spatial dimensions. In principle, the theory we developed predicts boundary-driven waves to be present in one-dimensional (*Figure 3*, *Figure 4*) and two-dimensional (*Figure 3F*) spatial systems. Therefore, we carried out experiments with droplets of cycling cell-free extracts on Teflon-coated glass slides, providing a thin structure, yet wide in diameter (≈ 1 mm). All such experiments showed that mitosis was

coordinated via mitotic waves that originate at the boundary, consistent with the theoretical predictions (see *Video 2*).

Finally, we analyzed each individual experiment in more detail with the goal to directly link the presence of pacemaker regions (be it at the boundary or internally) to a local increase in GFP-NLS intensity. This analysis confirmed that there is a higher build-up of GFP-NLS intensity towards the boundaries in wider tubes (see *Figure 5—figure supplement 4*).

Our findings illustrate that the spatial environment has a strong influence on how biological processes self-organize. In particular, increasing the spatial dimensions of the system leads to a higher probability of observing mitotic waves that originate at the boundary of the system. Other studies have also stressed the importance of system size, boundaries, and geometry on self-organization processes. For example, using cell-free frog extracts, cytoplasmic volume was demonstrated to determine the spindle size (*Good et al., 2013*; *Hazel et al., 2013*) and the size of the nucleus (*Hara and Merten, 2015*).

System boundaries (*Kopell et al., 1991*; *Haim et al., 1996*; *Rabinovitch et al., 2001*; *McNamara et al., 2016*; *Bernitt et al., 2017*) and system geometry (*Wettmann et al., 2018*) have been shown to affect the dynamics of traveling waves. In the widely studied amoeba *Dictyostelium discoideum*, the origin of cAMP waves have been studied in inhomogeneous systems. Waves appear spontaneously in areas of higher cell density with the oscillation frequency of these centers depending on their density (*Vidal-Henriquez and Gholami, 2019*). In the presence of advection, a boundary-induced instability was found to periodically excite a cAMP wave near the boundary (*Vidal-Henriquez et al., 2017*). Another well-characterized model organism is the bacterium *Escherichia coli*, where Min-protein wave patterns help select the site of cell division (*Hu and Lutkenhaus, 1999*; *Raskin and de Boer, 1999*). Wave patterns and the location of cell division have been shown to strongly depend on the system size and geometry, both *in vivo* by deforming cell shape (*Männik et al., 2012*; *Wu et al., 2015*; *Wettmann et al., 2018*) and *in vitro* by reconstituting Min oscillations in open and enclosed compartments (*Zieske and Schwille, 2014*; *Zieske et al., 2016*; *Caspi and Dekker, 2016*; *Wettmann et al., 2018*). As thin compartments were gradually increased in length, multiple regions of oscillations were observed (*Zieske and Schwille, 2014*; *Zieske et al., 2016*; *Caspi and Dekker, 2016*; *Wettmann et al., 2018*). For more complex geometries, many more wave patterns have been observed, such as standing waves, traveling planar and spiral waves, and coexisting stable stationary distributions (*Zieske and Schwille, 2014*; *Zieske et al., 2016*; *Caspi and Dekker, 2016*; *Wettmann et al., 2018*). While there are similarities with our findings in the *Xenopus* cell-free extracts, one important difference is that the wave patterns in the Min system are mainly controlled by the spatial dimensions and geometry. In contrast, in our findings the influence of the spatial dimensions are, at least partially, mediated by the nuclei within the oscillatory medium that serve as pacemakers.

## Discussion

A crucial task that a developing cell needs to accomplish is the replication of its DNA and, subsequently, cell division. In large cells, which demand spatial coordination in order to accomplish this task, mitotic waves can organize the process. We have demonstrated that nuclei act as pacemakers generating the mitotic waves in *Xenopus* cell-free extracts. Pacemakers are regions that oscillate faster than their environment, and, as such, initiate traveling waves (*Kuramoto, 1984*). A nucleus becomes a pacemaker by its ability to import factors into the nucleus and, presumably, concentrate cell cycle regulators. Indeed, we found that pacemakers are often located near nuclei that are brighter due to increased import of exogenously added GFP-NLS. We built a generic computational model, which showed that the distribution of cell cycle regulators also depends on the nuclear positioning and spatial dimensions of the system. We tested this idea by experimentally exploring the mitotic wave dynamics in cell-free extracts in which we changed the nuclear density and nuclear import strength. In cell-free extracts with only few nuclei, we found that mitotic waves originated at those nuclei and spread through the parts of the extract devoid of nuclei. In the absence of any nuclei in the system, no mitotic waves were observed. Decreasing the nuclear import strength similarly avoided the formation of mitotic waves. Finally, we changed the spatial dimensions of the system, and found that thicker tubes have a larger tendency to concentrate cell cycle regulators at the

boundaries, leading to mitotic waves originating at the outer edges of the tubes. Thus, nuclei are central hubs that organize this complex cellular process.

One advantage to having the nucleus control the timing of mitosis is that it allows the cell to ensure that DNA replication has completed before initiating mitosis. While DNA checkpoints are largely silenced in the early *Xenopus* embryo (*Newport and Dasso, 1989*), in *Drosophila* DNA content is known to activate the DNA-replication checkpoint and alter the cell cycle period (*Farrell and O'Farrell, 2014*; *Deneke et al., 2016*). A failure in the correct regulation of mitosis is associated with polyploidy, which plays a key role in nonmalignant physiological and pathological processes (*Fox and Duronio, 2013*). In the absence of a proper pacemaker, or if the pacemaker were to be located elsewhere, linking DNA replication to mitosis would be more complicated and, perhaps, more prone to error.

Previous studies have pointed to the critical role of the nucleus in spatial redistributing cell cycle regulators (*Gavet and Pines, 2010*; *Santos et al., 2012*). In particular, the nuclear import of Cyclin B has been shown to lead to spatial positive feedback, ensuring a robust and irreversible mitotic entry (*Santos et al., 2012*). Nuclei have also been found to be crucial in ensuring cell cycle oscillations in the *Drosophila* embryo (*Huang and Raff, 1999*; *Deneke et al., 2019*). Interestingly, although previous reports have suggested that centrosomes serve as pacemakers (*Chang and Ferrell, 2013*; *Ishihara et al., 2014*), we found that they are dispensable. After treating extracts with purified DNA, which lacks centrosomes, we still observed mitotic waves.

We also found that the interaction of multiple nuclei in a shared cytoplasm can lead to unexpected behavior. Nuclei self-organize in regular spatial patterns within a tube of *Xenopus* cell-free extract. The measured regular spacing between neighboring nuclei was found to be approximately 150 μm, which coincides with the nuclear subdomain of the cytoplasm in which spatial constraints show an effect on nuclear growth as studied in syncytial muscle fibers (*Landing et al., 1974*), *Drosophila* embryos (*Telley et al., 2012*), and cell-free frog extracts (*Hara and Merten, 2015*). It is also consistent with the size of cell-like compartments that spontaneously form in homogenized interphase cell-free frog extracts (*Cheng and Ferrell, 2019*). We found that such regularity in the nuclear distribution led to a build-up of cell cycle regulators towards the boundary of the system, such that the collective behavior of many nuclei creates a pacemaker region at the boundary of the oscillatory medium. This boundary effect was stronger with increasing widths of the tubes, in the presence of more extended regular nuclear patterns. We consistently observed more boundary-driven waves in such wider tubes.

Mitotic waves in the early *Drosophila* embryo also often originate at the boundary (*Foe and Alberts, 1983*). During nuclear cycles 10–13 in the syncytial blastoderm of these early embryos, nuclei enter (and exit) mitosis in waves that originate from the opposite anterior and posterior poles of the embryo and terminate in its mid-region. While mitotic waves are associated to so-called trigger waves in the *Xenopus* embryo (*Chang and Ferrell, 2013*; *Gelens et al., 2014*), they have been shown to be so-called sweep waves in the *Drosophila* embryo (*Vergassola et al., 2018*). We find, by computational modeling, that sweep waves are also able to generate boundary-driven waves in a syncytium, and that they propagate faster than trigger waves as predicted by *Vergassola et al. (2018)*; *Figure 4—figure supplement 2*. However, the internuclear distance of our simulations is significantly larger than the one observed in the more crowded *Drosophila* embryo, so it remains unclear whether our results can directly extend to that system. Despite the limitations of the model, our work is expected to be relevant for all coenocytes (*Ondracka et al., 2018*), where waves of mitosis have also been observed (*Sears, 1967*; *Brown et al., 2003*).

Nuclei are a natural choice of pacemaker for mitotic waves because they allow for a natural way to link one biological process, DNA replication, with another, mitosis. We hope that our work will further trigger new studies into the origin of pacemakers as the initiation of biological decisions mediated by traveling waves seem to be key in the proper coordination of a biological process. Traveling waves have, for example, also been found to propagate apoptosis (*Cheng and Ferrell, 2018*), action potentials (*Hodgkin and Huxley, 1952*), and calcium signals (*Stricker, 1999*) over large distances. In these systems, defective mitochondria, signals from neighboring neurons, or fertilization serve as the initial trigger to locally activate a wave.

# Materials and methods

## Key resources table

| Reagent type (species) or resource | Designation | Source or reference | Identifiers | Additional information |
|---|---|---|---|---|
| Strain, strain background (*Xenopus laevis*, male and female) | *Xenopus laevis* | Centre de Ressources Biologiques Xénopes | RRID:XEP_Xla | |
| Recombinant DNA reagent | GFP-NLS | DOI: 10.1038/nature12321 | | Construct provided by James Ferrell (Stanford Univ., USA) |
| Peptide, recombinant protein | (fluorescent) microtubule reporter | Cytoskeleton, Inc | Cat. #: TL488M-B | |
| Commercial assay or kit | GenElute Mammalian Genomic DNA kit | Sigma-Aldrich | Cat. #: G1N70 | |
| Chemical compound, drug | Human chorionic gonadotropin | MSD Animal Health | | CHORULON |
| Chemical compound, drug | Pregnant mare's serumgonadotropin | MSD Animal Health | | FOLLIGON |
| Chemical compound, drug | Calcium ionophore A23187 | Sigma-Aldrich | PubChem CID: 11957499; Cat. #: C7522 | |
| Chemical compound, drug | Leupeptin | Sigma-Aldrich | PubChem CID: 72429; Cat. #: L8511 | |
| Chemical compound, drug | Pepstatin | Sigma-Aldrich | PubChem CID: 5478883; Cat. #: P5318 | |
| Chemical compound, drug | Chymostatin | Sigma-Aldrich | PubChem CID: 443119; Cat. #: C7268 | |
| Chemical compound, drug | Cytochalasin B | Sigma-Aldrich | PubChem CID: 5311281; Cat. #: C6762 | |
| Chemical compound, drug | Proteinase K | Sigma-Aldrich | Cat. #: P2308 | |
| Chemical compound, drug | Importazole | Sigma-Aldrich | PubChem CID: 2949965; Cat. #: SML0341 | |
| Chemical compound, drug | S-Trityl-L-cysteine | Acros Organics | PubChem CID: 76044; Cat. #: 173010050 | |
| Software, algorithm | Fiji | http://fiji.sc/ | RRID:SCR_002285 | |
| Software, algorithm | Wolfram Mathematica | www.wolfram.com/mathematical | RRID:SCR_014448 | |
| Software, algorithm | Ilastik | www.ilastik.org | RRID:SCR_015246 | |
| Software, algorithm | Model for nuclear import | This paper, used for *Figure 3* | | Code on GitHub (*Nolet, 2020*) |
| Software, algorithm | Model for nuclear import, frequency dependent | This paper, used for *Figure 4* | | Code on GitHub (*Nolet, 2020*) |
| Other | Teflon tube | Cole-Parmer | Cat. #: 06417–11 | |

*Continued on next page*

Continued

| Reagent type (species) or resource | Designation | Source or reference | Identifiers | Additional information |
|---|---|---|---|---|
| Other | Hoechst 33342 | ImmunoChemistry technologies | RRID:AB_265113; Cat. #: 639 | (5 μg/mL) |
| Other | Leica TCS SPE confocal microscope | Leica Microsystems | RRID:SCR_002140 | |
| Other | Ultracentrifuge OPTIMA XPN - 90 | Beckman Coulter | RRID:SCR_018238; Cat. #: A94468 | |

## Numerical integration

All PDE models are solved by numerical integration using custom-made Fortran scripts. Discretization in time is done with a forward Euler method, while discretization in space is carried out with a central difference method. Data is written to `.txt` files which are then analyzed in Mathematica. The ODE models (CCO and FHN for *Figure 4—figure supplement 1*) are directly solved in Mathematica, since computational time is limited to seconds. The numerical codes that were used are available through GitHub (*Nolet, 2020*).

## Experimental setup

We reconstitute cell cycle oscillations *in vitro* in cell-free cycling extracts made from unfertilized *Xenopus laevis* frog eggs, following the protocol by *Murray, 1991*; *Box 2*). Female *Xenopus laevis* frogs are injected subcutaneously with 500 injection units (IU) human chorionic gonadotropin (MSD Animal Health) to induce ovulation, after prior priming with 100 IU pregnant mare's serum gonadotropin (MSD Animal Health). The obtained eggs are rinsed with deionized water and subsequently their jelly coat is removed by incubation in a 2% w/v cysteine in $1 \times$ XB salts solution. Dejellied eggs are now susceptible to activation with the calcium ionophore A23187 (0.5 μg/mL in $0.2 \times$ Marc's Modified Ringer's buffer, Sigma-Aldrich) for 2 min to start the biochemical processes of the cell cycle. After a packing step, the activated eggs are crushed in an ultracentrifuge (XPN90, Optima) at $16,000 \times g$ at 2°C for 10 min. This allows the collection of the cytoplasmic fraction to which the protease inhibitors leupeptin, pepstatin and chymostatin (Sigma-Aldrich) are added to a final concentration of 10 μg/mL. Cytochalasin B (10 μg/mL, Sigma-Aldrich) is also added to inhibit actin assembly and thus gelation-contraction, keeping the extract fluid at room temperature (*Field et al., 2011*).

Finally, the extract is supplemented with GFP-NLS (∼ 25 μM), green fluorescent protein with a nuclear localization signal, and sperm chromatin (using two different concentrations: ∼ 63 or 250 nuclei/μL extract). The construct for GFP-NLS was kindly provided by James Ferrell (Stanford Univ., USA). Sperm chromatin was prepared according the protocol by *Murray, 1991*. The supplemented extracts are then loaded in Teflon tubes (Cole-Parmer PTFE, 06417–11), through aspiration, and imaged at 24°C on a Leica TCS SPE confocal fluorescence microscope. This approach allows to visualize regular oscillations between interphase and mitotic phase. In interphase, nuclei form spontaneously in the extract supplemented with sperm chromatin. These nuclei then import GFP-NLS (see *Box 2*). In mitosis, the nuclear envelope breaks down and GFP is no longer localized to nuclei. Here, we use this experimental system to explore the influence of system size by varying the width of the Teflon tubes. The tubes were approximately 100, 200, 300, and 560 μm in width (the actual inner diameters are 102, 203, 305, and 559 μm). Furthermore, we change the amount of nuclear material and its distribution by considering two different concentrations of added sperm chromatin.

In addition, DNA was purified from the sperm chromatin. This was done using a GenElute Mammalian Genomic DNA kit (Sigma-Aldrich), with the use of proteinase K (Sigma-Aldrich) to release the DNA from the histones and give a higher yield. After purification, the concentration of DNA was determined using a NanoDrop spectrophotometer. Purified DNA was added to the extract at final concentrations of 5, 10, 15, 20, 25, 45 and 60 ng/μL.

Nuclear import was inhibited by adding importazole (Sigma-Aldrich), an inhibitor of importin-$\beta$ transport receptors. Final concentrations of 5, 10, 20, 40, and 60 μM were tested.

Microtubule dynamics was disrupted by adding *S*-Trityl-L-cysteine (STLC, Acros Organics), a kinesin Eg5 inhibitor. Final concentrations of 10, 20, 30, 40, and 50 μM were tested.

In some of the experiments fluorescent reporters other than GFP-NLS were used. These included a green microtubule reporter (Tubulin porcine HiLyte 488; Cytoskeleton, Inc) at 1 μM final concentration and DNA staining (Hoechst 33342) at 5 μg/mL final concentration.

## Image analysis

### Microscope data

We used a Leica TCS SPE confocal fluorescence microscope (5x objective) in confocal mode to excite the GFP-NLS with a 488 nm solid state laser, and capture the emission from 493 to 600 nm. In the non-confocal experiments we used the Leica EL6000 metal halide external fluorescence light source for excitation of the fluorophores. The different filter cubes used were the L5 (excitation 480/40 nm bandpass, emission 527/30 nm bandpass) for GFP-NLS and HiLyte Fluor 488; and the A4 (excitation 360/40 nm bandpass, emission 470/40 nm bandpass) for the Hoechst 33342 staining. First, we fixed imaging positions at different (x,y) locations of the Teflon tubes, ensuring overlap between subsequent positions to capture the whole tubes. Within a tube, the z-position was fixed, but could differ between tubes to be able to image the central plane of the tubes. We then captured time-lapse images of these different positions during 18 hr, creating image stacks for each position in a `.lif` (Leica Image File) format. The `.lif` files belonging to one tube were then imported in Fiji (*Schindelin et al., 2012*). The maximum intensity of the different image stacks was put at the same level. Then, using the overlap between subsequent image positions, the image stacks were stitched pairwise (*Preibisch et al., 2009*). Subsequently, the images were cropped and saved as separate `.tiff` files per timepoint, an `.avi` file and a kymograph were made.

### Data analysis from images

The `.tiff` files are imported in Mathematica and for all $x$ the maximum intensity over the width is calculated. This allows us to have a one-dimensional intensity profile for each time, see *Figure 1—figure supplement 1C*. Kymographs as in *Figure 1A* and *Figure 1—figure supplement 3* were made from these profiles over time. Lines are drawn through the points of mitotic entry (disappearance of nuclei), for every visible cycle. This is done by manually detecting the start- and endpoints of the wave, as depicted in the sketch of *Figure 1—figure supplement 1D*. The lines are drawn through those points automatically and periods and wave speeds are then calculated based on these lines. The period is calculated by taking 20 points on these lines and determining the time to the next line. This gives an average period (and standard deviation) for each cycle. The wave speed is calculated by taking the derivative of the lines. For the full cycle, the wave speed is only reported if the wave travels a large enough (> 600 μm) distance (to only include well-formed waves and to reduce noise), and if multiple waves are present, the minimum speed is reported. The locations of the nuclei (one-dimensional) are extracted from the kymographs at the last one or two lines (if nuclei are well-separated). For each nucleus the average distance to their neighbors (left and right) is calculated which is also plotted in *Figure 1C* and *Figure 1—figure supplement 3*. For the last two cycles, the maximum intensity over the cycle is calculated at every $x$, yielding an intensity profile at each cycle.

### Processing for specific analyses

When calculating properties of individual nuclei (e.g. size, location, intensity), the Ilastik software was used to automatically recognize nuclei in a series of `.tiff` files. This program relies on machine learning software which makes recognition a lot faster than manual tracking. The files are imported in Ilastik, where we provided three labels ('nucleus', 'background' or 'outside of the tube') to train the implemented random forest classifier to recognize the labels in the images (*Sommer et al., 2011*). After the training phase, we exported the results as a `.hdf5` file, which contains the probability of each pixel to be 'nucleus', 'background' and 'outside of the tube' for each timepoint. The .hdf5 files were imported in Mathematica for further analysis. The data of these files was binarized by defining all pixels with a high probability ($\geq$ 75%) as nuclei (1) and others as background (0). Adjacent pixels were grouped together and the separate groups were recognized as the nuclei. Noise was reduced by ignoring nuclei consisting only of a few pixels. This resulted in a binarized picture, such as in *Figure 1—figure supplement 1A*. Of all recognized nuclei (orange), information as location (center) and size is extracted with Mathematica. In order to obtain continuous-time kymographs

(such as in *Figure 1A* and *Figure 5—figure supplement 1*, we overlayed the binarized matrix with the original `.tiff` and integrated over the width. In this way intensity differences were still visible.

## Analysis of the pacemaker strength of internal regions and the boundary regions

The GFP-NLS intensity profile of the experiments is analyzed in order to calculate the strength of the boundary and of internal pacemakers (*Figure 5—figure supplement 1* and *Figure 5—figure supplement 4*). An example of such an intensity profile $I(x)$ is shown in *Figure 1—figure supplement 1B*. The averaged intensity profile is filtered using a low-pass filter, to obtain a 'background' signal $y(x)$. This is the red line in *Figure 1—figure supplement 1B*. All frequencies higher than a threshold $s>0$ are filtered out. The obtained background profile $y(x)$ does of course depend on the parameter $s$. The position of the minimum of $y(x)$ is denoted by $\bar{x}$, that is

$$y(\bar{x}) = \min_{x \in [0,L]} y(x). \tag{1}$$

From the background profile, we calculate two measures $L_1, R_1$ for the GFP build-up at the boundary, by

$$L_1 = \frac{1}{\bar{x}} \int_0^{\bar{x}} (y(x) - y(\bar{x})) dx \tag{2}$$

and

$$R_1 = \frac{1}{L - \bar{x}} \int_{\bar{x}}^L (y(x) - y(\bar{x})) dx. \tag{3}$$

These correspond to the GFP build-up in the blue areas in *Figure 5—figure supplement 1*.

A second parameter, $k>0$, is introduced and defines the boundary width. In other words, the intervals $[0,k]$ and $[L-k,L]$ are the boundary domains and $[k,L-k]$ is the internal domain. The background profile $y(x)$ might over- or underestimate GFP build-up in the boundary domains. This is compensated by calculating the second type of measures, $L_2$ and $R_2$. These are defined by

$$L_2 = \frac{1}{k} \int_0^k (I(x) - y(x)) dx \tag{4}$$

and

$$R_2 = \frac{1}{k} \int_{L-k}^L (I(x) - y(x)) dx. \tag{5}$$

The GFP build-up at the boundary, denoted by $\Gamma_b$, of this intensity profile is now defined as

$$\Gamma_b = \max\{L_1 + L_2, R_1 + R_2\}. \tag{6}$$

The internal GFP build-up (i.e. by nuclei located internally) is defined by those areas where the intensity $I(x)$ is higher than the background profile $y(x)$. This internal GFP build-up $\Gamma_i$ is calculated by

$$\Gamma_i = \frac{1}{L - 2k} \int_k^{L-k} \max\{0, I(x) - y(x)\} dx, \tag{7}$$

which correspond to the orange areas in *Figure 5—figure supplement 1*.

*Figure 5—figure supplement 4* shows the GFP build-up at the boundary and internally, $\Gamma_i$ vs. $\Gamma_b$, for 20 experiments. This is done for various values of $k$ and $s$. Since $\Gamma_i$ and $\Gamma_b$ depend on these parameters, the figure will change with those parameters. However, we see that qualitatively differences are small.

## Data availability

All the data generated during the study are summarized and provided in the manuscript and supporting files. Source files have been provided for *Figure 1*, *Figure 1—figure supplement 4*,

*Figure 2*, *Figure 5—figure supplement 1*, *Box 2*, *Video 1* and *Video 2* in the format of microscopy videos. Additionally, representative microscopy videos of all different conditions are provided as a Zenodo dataset (http://doi.org/10.5281/zenodo.3736728). The numerical codes that were used, together with an overview table of the performed experiments, are available through GitHub (*Nolet, 2020*; copy archived at https://github.com/elifesciences-publications/eLife_paper).

## Acknowledgements

We thank Jim Ferrell, Sophie De Buyl, and Jan Rombouts for valuable feedback on the manuscript. We also thank Jonás Noguera López, Ine Vlaeminck and Virginia Tsiouri for their help in the lab and stitching movies. This work was supported by the Research Foundation - Flanders (grant GOA5317N) and the KU Leuven Research Fund (C14/18/084).

## Additional information

### Funding

| Funder | Grant reference number | Author |
|---|---|---|
| Research Foundation - Flanders | GOA5317N | Lendert Gelens |
| KU Leuven Research Fund | C14/18/084 | Lendert Gelens |

The funders had no role in study design, data collection and interpretation, or the decision to submit the work for publication.

### Author contributions

Felix E Nolet, Data curation, Software, Formal analysis, Validation, Investigation, Visualization, Methodology, Writing - review and editing, Carried out the modeling and simulation work, and the analysis of the experimental data; Alexandra Vandervelde, Validation, Investigation, Methodology, Writing - review and editing, Set up and carried out most of the experiments for the first submission; Arno Vanderbeke, Data curation, Validation, Investigation, Methodology, Writing - review and editing, Carried out part of the experiments, in particular for the revisions; Liliana Piñeros, Data curation, Investigation, Methodology, Writing - review and editing, Carried out part of the experiments, in particular for the revisions; Jeremy B Chang, Conceptualization, Validation, Investigation, Methodology, Writing - review and editing, Initial experimental work with droplets of cell-free extracts, which helped trigger this project; Lendert Gelens, Conceptualization, Resources, Supervision, Funding acquisition, Visualization, Methodology, Writing - original draft, Project administration, Writing - review and editing

### Author ORCIDs

Felix E Nolet https://orcid.org/0000-0001-9300-6302
Arno Vanderbeke https://orcid.org/0000-0002-7240-8377
Liliana Piñeros https://orcid.org/0000-0001-8872-9602
Jeremy B Chang https://orcid.org/0000-0002-7381-6444
Lendert Gelens https://orcid.org/0000-0001-7290-9561

### Ethics

Animal experimentation: This study was performed in strict accordance with the recommendations in the Guide for the Care and Use of Laboratory Animals of the KU Leuven. All of the animals were handled according to approved institutional animal care and use committee (IACUC) protocols of the KU Leuven. The protocol was approved by the Committee on the Ethics of Animal Experiments of the KU Leuven (ECD permit Number: P165/2016/).

### Decision letter and Author response
Decision letter https://doi.org/10.7554/eLife.52868.sa1
Author response https://doi.org/10.7554/eLife.52868.sa2

## Additional files

### Supplementary files

- Transparent reporting form

### Data availability

All the data generated during the study are summarized and provided in the manuscript and supporting files. Source files have been provided for Figure 1, Figure 1-figure supplement 3, Figure 2, Figure 5-figure supplement 1, Box 2, Video 1 and Video 2 in the format of microscopy videos. Additionally, representative microscopy videos of all different conditions are provided as a Zenodo dataset (http://doi.org/10.5281/zenodo.3736728). The numerical codes that were used, together with an overview table of the performed experiments, are available through GitHub (https://github.com/felixnolet/eLife_paper; copy archived at https://github.com/elifesciences-publications/eLife_paper).

The following dataset was generated:

| Author(s) | Year | Dataset title | Dataset URL | Database and Identifier |
|---|---|---|---|---|
| Nolet FE, Vandervelde A, Vanderbeke A, Pineros L, Chang JB, Gelens L | 2020 | Nuclei determine the spatial origin of mitotic waves | http://doi.org/10.5281/zenodo.3736728 | Zenodo, 10.5281/zenodo.3736728 |

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

## Appendix 1

### Modeling import and diffusion processes

We have discussed two models of import and diffusion of cell cycle regulators and have shown their results in **Figure 3** and **Figure 4**. Here we elaborate on the model specifics.

### A model of import and diffusion of cell cycle regulators with a fixed cell cycle period

We implemented the competing import and diffusion processes through a simple PDE equation that describes the evolution of the concentration $C$ of a nuclear protein:

$$\frac{\partial C}{\partial t} = D\nabla^2 C + F(t)[\nabla \cdot (C\nabla V)] \tag{8}$$

with $D$ a diffusion constant, $F(t)$ a periodic function that periodically switches between 0 (in mitosis) and 1 (in interphase),

$$F(t) = \begin{cases} 1 & \text{if } t \bmod T < \alpha T \\ 0 & \text{if } t \bmod T \geq \alpha T \end{cases} \tag{9}$$

with $T$ the cell cycle period, $\alpha$ the ratio of interphase to the cell cycle period and $V = V(\vec{x})$ a potential function to define nuclear attraction at different locations. For simplicity, we use a sum of Gaussian potential wells throughout this work,

$$V(\vec{x}) = -\epsilon \sum_{i=1}^{N} \frac{1}{\sigma\sqrt{2\pi}} e^{-\frac{1}{2\sigma^2}(\vec{x}-\vec{\xi}_i)\cdot(\vec{x}-\vec{\xi}_i)} \tag{10}$$

with $N$ attracting positions at $\vec{\xi}_i$ (the 'nuclei'), and $\epsilon, \sigma > 0$ parameters that define the amplitude and the width of the potential wells, respectively. The analysis is also repeated with a different potential function, giving qualitatively similar results.

Rewriting the one-dimensional model equation as

$$\frac{\partial C}{\partial t} = D\frac{\partial^2 C}{\partial x^2} + F(t)\left(\frac{\partial C}{\partial x}\frac{\partial V}{\partial x} + C\frac{\partial^2 V}{\partial x^2}\right) \tag{11}$$

we decrease numerical errors by explicitly implementing the first and second derivative of $V$. With basic calculus we find for a Gaussian potential function,

$$\frac{\partial V}{\partial x} = \epsilon \sum_{i=1}^{N} \frac{x-\xi_i}{\sigma^3\sqrt{2\pi}} e^{-\frac{1}{2\sigma^2}(x-\xi_i)^2} \tag{12}$$

and

$$\frac{\partial^2 V}{\partial x^2} = \epsilon \sum_{i=1}^{N} \left(\frac{1}{\sigma^3\sqrt{2\pi}} e^{-\frac{1}{2\sigma^2}(x-\xi_i)^2} - \frac{(x-\xi_i)^2}{\sigma^5\sqrt{2\pi}} e^{-\frac{1}{2\sigma^2}(x-\xi_i)^2}\right). \tag{13}$$

For the appropriate boundary conditions we need to calculate the flux. Let $\vec{J}$ be the flux of $C$, then we have

$$\frac{\partial C}{\partial t} + \nabla \cdot \vec{J} = 0 \tag{14}$$

and when rewriting the model equation as

$$\frac{\partial C}{\partial t} = \nabla \cdot (D\nabla C + F(t)C\nabla V) \tag{15}$$

we find the flux to be

$$\vec{J} = -(D\nabla C + F(t)C\nabla V) \tag{16}$$

and the zero-flux boundary condition is then $\vec{J} \cdot \hat{n} = 0$, with $\hat{n}$ the normal vector at the boundary. In one dimension we will have a boundary condition at $x = 0$ given by

$$D\frac{\partial C}{\partial x}\big|_{x=0} + F(t)C(0,t)\frac{\partial V}{\partial x}\big|_{x=0} = 0 \tag{17}$$

for all $t$ and for the numerical derivative of $C$ this becomes

$$D\frac{C(\Delta x, t) - C(0,t)}{\Delta x} + F(t)C(0,t)\frac{\partial V}{\partial x}\big|_{x=0} = 0 \tag{18}$$

where $\Delta x$ denotes the numerical grid distance. This yields

$$C(0,t) = \frac{DC(\Delta x, t)}{D - \Delta x F(t)\frac{\partial V}{\partial x}\big|_{x=0}} \tag{19}$$

for which we use the explicit formula for $\frac{\partial V}{\partial x}$. Similarly, one can find at $x = L$,

$$C(L,t) = \frac{DC(L - \Delta x, t)}{D + \Delta x F(t)\frac{\partial V}{\partial x}\big|_{x=L}}. \tag{20}$$

## A model of import and diffusion of cell cycle regulators with a concentration-dependent cell cycle period

The previous model is uniform in time, that is the cell cycle is regulated by the function $F(t)$ with a given length of S phase and M phase. Here we let the period of each nucleus depend on the local concentration of cell cycle regulators. This is done by including a separate function $F$ in the potential for every nucleus and letting the period depend on $C$. This means the period $T$ is not a fixed constant anymore, every nucleus has its own $T$ that depends on $C$ (and therefore on time).

The model equation becomes

$$\frac{\partial C}{\partial t} = D\nabla^2 C + \nabla \cdot (C\nabla V) \tag{21}$$

and recall that every nucleus defines a potential $V_i(\vec{x}, t)$ and together they form the full potential function

$$V(\vec{x}, t) = \sum_{i=1}^{N} V_i(\vec{x}, t) \tag{22}$$

via superposition. The single-nucleus potential can now be written as

$$V_i(\vec{x}, t) = F_i(t)G_i(\vec{x}), \tag{23}$$

i.e. it can be separated in a time-dependent and a space-dependent function. The function $G_i$ is the same Gaussian well as before, that is

$$G_i(\vec{x}) = -\beta_i \frac{\epsilon}{\sigma\sqrt{2\pi}} e^{-\frac{1}{2\sigma^2}(\vec{x} - \vec{\xi}_i) \cdot (\vec{x} - \vec{\xi}_i)} \tag{24}$$

where $\sigma > 0$ is a parameter that defines the width of the potential function (i.e. the attraction range of the nucleus) and $\epsilon > 0$ is the amplitude of the potential (i.e. the

attraction strength). The parameter $\beta_i$ is nucleus-dependent and can be used to give different strengths to different nuclei. Normally, when all nuclei have the same attraction strength, $\beta_i = 1$ for all $i$. The function $F_i$ regulates the potential over time, essentially by turning it on and off. The definition is

$$F_i(t) = \begin{cases} 1 & \text{if } t - t_0 < \alpha T_i(t) \text{ and } t - t_1 \geq (1-\alpha)T_i(t) \\ 0 & \text{if } t - t_0 \geq \alpha T_i(t) \text{ and } t - t_1 < (1-\alpha)T_i(t) \end{cases} \tag{25}$$

where $t_0$ and $t_1$ are time-dependent via

$$t_k = \max\{s \in [0,t] | F_i(s) = k\} \tag{26}$$

for $k \in \{0,1\}$, that is $t_0$ denotes the last $t$ for which $F_i(t) = 0$ and $t_1$ the last $t$ for which $F_i(t) = 1$. Therefore the function depends on its history and we define $F_i(0) = 1$. The parameter $0 < \alpha < 1$ determines the fraction of the cycle where nuclei are in attracting phase. The length of the cycle, denoted by the period function $T_i$, is not constant but changes in time. This is because it is linked to the concentration at the position $\vec{\xi}_i$ of the nucleus via

$$T_i(t) = \frac{T}{1 + \gamma(C(\vec{\xi}_i, t) - C_0)} \tag{27}$$

where $T$ is a reference period (chosen the same as in the previous model), $C_0$ a reference concentration and $\gamma \in \mathbb{R}$ is a parameter that determines how strong the period is coupled to the concentration at $\vec{\xi}_i$. This time-dependent period $T_i$ is concentration-dependent (which on itself is time-dependent) and has as a consequence that nuclei with a higher concentration have a shorter period (for $\gamma > 0$). The underlying biological assumption is that nuclei that import (attract) more cell cycle regulators have a shorter cell cycle length than other nuclei.

Boundary conditions are the same as before. The standard values of the model parameters are given in the table below.

| Par. | Value | Unit |
|---|---|---|
| D | 600 | $\mu m^2/min$ |
| T | 40 | min |
| $\alpha$ | 0.7 | |
| $\beta_i$ | 1 | |
| $\gamma$ | 1 | |
| $C_0$ | 1.1 | |
| $\epsilon$ | $4 \cdot 10^4$ | $\mu m^3/min$ |
| $\sigma$ | 60 | $\mu m$ |

## Appendix 2

# Cell cycle models

## Biochemical cell cycle oscillator

Mathematically, the cell cycle oscillator (CCO) based on the regulatory network in *Figure 4—figure supplement 1A–C* can be described using the following ODEs for cyclin B concentrations ([cyc]) and concentrations of the cyclin B-Cdk1 complex in its active state ([cdk1]),

$$
\begin{cases}
\frac{d[\text{cyc}]}{dt} = -\left(a_3 + b_3 \frac{[\text{cdk1}]^{n_3}}{E_3^{n_3} + [\text{cdk1}]^{n_3}}\right)[\text{cyc}] + k \\
\frac{d[\text{cdk1}]}{dt} = \left(a_1 + b_1 \frac{[\text{cdk1}]^{n_1}}{E_1^{n_1} + [\text{cdk1}]^{n_1}}\right)([\text{cyc}] - [\text{cdk1}]) \\
\qquad - \left(a_2 + b_2 \frac{E_2^{n_2}}{E_2^{n_2} + [\text{cdk1}]^{n_2}}\right)[\text{cdk1}] \\
\qquad - \left(a_3 + b_3 \frac{[\text{cdk1}]^{n_3}}{E_3^{n_3} + [\text{cdk1}]^{n_3}}\right)[\text{cdk1}] + k
\end{cases}
$$

with $k$ the synthesis rate of cyclin B, and $a_i, b_i, E_i, n_i$ parameters corresponding to the influence of Cdc25 ($i = 1$), Wee1 ($i = 2$) and degradation via APC/C ($i = 3$). For a more detailed discussion of this model, including an experimental motivation and/or measurement of the different parameter values, we refer to *Chang and Ferrell, 2013* and references therein.

## Generic FitzHugh-Nagumo oscillator

Although the CCO model (*Equation (28)*) includes known biochemical interactions, much of the mitotic wave dynamics solely results from the fact that the cell cycle is driven by a relaxation-type oscillator, such as the generic FHN oscillator (*Fitzhugh, 1961*). The FHN system was originally constructed as a simplified version of the Hodgkin-Huxley model, describing the activation and deactivation dynamics of a spiking neuron (*Hodgkin and Huxley, 1952*). Similarly, the FHN model has later been used to describe pulse dynamics in the heart (*Aliev and Panfilov, 1996*) and cell cycle oscillations (*Gelens et al., 2014*; *Gelens et al., 2015*). Using such a simple model to describe complex biological processes has the advantage of being numerically more effective, allowing for analytical estimations, and having fewer parameters.

We use a generalized form of the classical FHN oscillator, including two ODEs for variables $u$ and $v$,

$$
\begin{cases}
\frac{du}{dt} = -u^3 + cu^2 + du - v \\
\frac{dv}{dt} = \epsilon(u - bv + a)
\end{cases}
\tag{29}
$$

with parameters $a, b, c, d \in \mathbb{R}$ and $\epsilon > 0$. When taking $a = c = 0$, the original FHN equations are recovered. *Figure 4—figure supplement 1E,F* show that the relaxation oscillations in the FHN are indeed very similar to the cell cycle oscillations, after applying a linear mapping $(u, v, t) \mapsto (-0.19u + 0.5, 0.32v + 0.52, 5.75t)$ (chosen by trial-and-error to obtain visual correspondence).

## Spatially extended models

In order to capture spatial dynamics, we extended the CCO model (*Equation (28)*) and FHN model (*Equation (29)*) by including free diffusion of proteins (with diffusion constants $D \sim 10 - 1000\ \mu\text{m}^2/\text{min}$), leading to the following coupled partial differential equations (PDEs) for the CCO and FHN system, respectively:

$$\begin{cases} \frac{\partial [\text{cyc}]}{\partial t} = D\nabla^2[\text{cyc}] \\ \qquad -(a_3 + b_3 \frac{[\text{cdk1}]^{n_3}}{E_3^{n_3} + [\text{cdk1}]^{n_3}})[\text{cyc}] + k, \\ \frac{\partial [\text{cdk1}]}{\partial t} = D\nabla^2[\text{cdk1}] \\ \qquad +(a_1 + b_1 \frac{[\text{cdk1}]^{n_1}}{E_1^{n_1} + [\text{cdk1}]^{n_1}})([\text{cyc}] - [\text{cdk1}]) \\ \qquad -(a_2 + b_2 \frac{E_2^{n_2}}{E_2^{n_2} + [\text{cdk1}]^{n_2}})[\text{cdk1}] \\ \qquad -(a_3 + b_3 \frac{[\text{cdk1}]^{n_3}}{E_3^{n_3} + [\text{cdk1}]^{n_3}})[\text{cdk1}] + k, \end{cases}$$

and

$$\begin{cases} \frac{\partial u}{\partial t} = D\nabla^2 u - u^3 + cu^2 + du - v, \\ \frac{\partial v}{\partial t} = D\nabla^2 v + \epsilon(u - bv + a). \end{cases} \tag{31}$$

## Parameter values

Standard parameters of both the CCO (top) and FHN (bottom) models are given in the table below. When other values are used for certain figures, this is stated in the corresponding caption. Note that since the FHN is a dimensionless model, no units are reported.

| Parameter | Value | Unit | Parameter | Value | Unit |
|---|---|---|---|---|---|
| $a_1$ | 0.8 | $\text{min}^{-1}$ | $b_1$ | 4 | $\text{min}^{-1}$ |
| $a_2$ | 0.4 | $\text{min}^{-1}$ | $b_2$ | 2 | $\text{min}^{-1}$ |
| $a_3$ | 0.01 | $\text{min}^{-1}$ | $b_3$ | 0.06 | $\text{min}^{-1}$ |
| $E_1$ | 35 | nM | $n_1$ | 11 | |
| $E_2$ | 30 | nM | $n_2$ | 3.5 | |
| $E_3$ | 32 | nM | $n_3$ | 17 | |
| $k$ | 1.5 | nM/min | $D$ | 600 | $\mu\text{m}^2/\text{min}$ |
| $a$ | −0.85 | | $b$ | 0.05 | |
| $c$ | 1.2 | | $d$ | 0.5 | |
| $\epsilon$ | 0.01 | | $D$ | 600 | |

## Modeling sweep waves in the FHN model

Sweep waves are caused by moving the nullcline in time, which increases the frequency and can also increase the wave speed in an already oscillating model. In the FHN model, this is implemented by adding an extra term, $\beta$, as follows:

$$\begin{cases} \frac{\partial u}{\partial t} = D\nabla^2 u - u^3 + cu^2 + du - v - \beta, \\ \frac{\partial v}{\partial t} = D\nabla^2 v + \epsilon(u - bv + a). \end{cases} \tag{32}$$

This term in the $u$-equation moves the nullcline (***Figure 4—figure supplement 1F***) to the left for positive $\beta$. The term depends on time: it is zero in M phase and increases in S phase. We distinguish two types of sweep waves, local and global. In the local case, $\beta$ is altered at every $x$, that is $\beta = \beta(x,t)$. In the global case, $\beta$ only depends on time but is uniform in space. The increase in S phase is with a certain speed, $v_{\text{sweep}}$ and $\beta$ then takes the form $\beta = \frac{1}{30}v_{\text{sweep}}t$. The fraction 1/30 is for convenience (since the duration of S phase is around 30 min one can easily check how far the nullcline is swept).

