## [Decision Letter]

Thank you for submitting your article "Nuclei and spatial dimensions determine pacemaker location in mitotic waves" for consideration by *eLife*. Your article has been reviewed by three peer reviewers, one of whom is a member of our Board of Reviewing Editors, and the evaluation has been overseen by Naama Barkai as the Senior Editor. The following individual involved in review of your submission has agreed to reveal their identity: William M Bement (Reviewer #2).

The reviewers have discussed the reviews with one another and the Reviewing Editor has drafted this decision to help you prepare a revised submission.

Summary:

The manuscript by Nolet et al. addresses an interesting scientific question, i.e. what sets the location where Cdk1 signaling waves start. Using *Xenopus* extracts, the authors propose that the efficiency of nuclear import and the geometry of the system are major parameters controlling the position where waves start. These insights are in principle interesting. However, there are significant concerns that must be addressed before the paper can be re-evaluated.

Essential revisions:

1) Biological Significance. It is unclear what the biological significance of the presented results is. The proposal that the experiments will apply to *Drosophila* embryos is complicated by several observations in that system that argue for significant differences from your results: a) Cdk1 waves are not trigger waves in fly embryos (Vergassola et al., 2018); b) the correlation length of the field of Cdk1 activity (~100 microns) is significantly larger than internuclear distance (<10 micron) suggesting that nuclei are unlikely to act in isolation in that system; c) the local nuclear-to-cytoplasmic ratio is important for timing the cell cycle. Thus, we recommend that you propose other possible *in vivo* applications of your insights from *in vitro* experiments.

2) Evidence for nuclei acting as pacemakers. The evidence that nuclei act as a pacemaker should be strengthened. There are several suggestions from the reviewers. While answering all the criticism might not be necessary, we would like you to address the main concerns of reviewer 2, as well as provide evidence that nuclear brightness is a better predictor of wave initiation than a measurement of the nuclear-to-cytoplasmic ratio (for example you could divide the extract in cell-like domains by some geometric tessellation and check whether nuclear brightness is a better predictor of wave initiation than the size of the cell-like domain). While FRAP experiments would be a great quantitative test of nuclear import/export parameters, they are not necessary if you lack the setup for performing them.

3) Theoretical analysis. Reviewer 3 raises several theoretical questions that should be dealt with.

*Reviewer #1:*

Signaling waves are emerging as a general mechanism of regulation of biological processes. Among them, mitotic waves in early embryos are particularly interesting as they are both functionally important and a great system for quantitative dissection of the mechanisms of the waves. The paper by Nolet and collaborators addresses the mechanisms by which certain regions can emerge as pacemakers in large spatial systems. The authors use the *Xenopus* extract system which is able to undergo repeated cell cycles which organize in mitotic waves. The authors argue for two major conclusions: waves tend to start at nuclei that are more efficient at nuclear import; spatial dimensions influence whether waves tend to start near the boundary or in the middle of the domain. These are in principle interesting observations, but this paper suffers from significant limitations that need to be addressed. Specifically the significance of their *in vitro* findings for *in vivo* systems is unclear and the experimental analysis could be strengthen by more quantifications and additional experiments.

1) While the experiments presented in this paper are well-executed, their relevance to *in vivo* systems remain unclear. The authors speculate that their results could explain the origin of waves in *Drosophila* embryos. However, this proposal is problematic and it is important that the authors address the fact that mitotic waves in *Drosophila* are not trigger waves (Vergassola et al., 2018). The authors need to study wave origin in the context of the mechanism proposed by Vergassola et al. if they wanted to make claims about the *Drosophila* embryo. Also notice that the inter-nuclear distances in these two systems are significantly different. Overall, I am very skeptical that one can generalize the findings of this paper to the fly embryo.

2) The authors make the claim on more efficient nuclear import based on nuclear fluorescent intensity. However, it is my impression that many of the nuclei in the system do not separate following mitosis suggesting that these nuclei might ultimately become polyploid. So, I find nuclear fluorescence and intensity difficult to interpret. A better test of the nuclear import efficiency would be to use FRAP to measure the kinetic parameters of nuclear import and export. It would also be important to test how large the difference in nuclear import need to be to impact cell cycle durations. It is unclear how physiological nuclear dynamics is in the egg extract. Finally, it is in principle possible that the cell cycle influences nuclear import and vice versa. This would leave a bit of chicken/egg problem. Is nuclear import causing faster cell cycles or is higher nuclear import a consequence of the faster cell cycles? This paper presents no attempt at manipulating nuclear inputs or cell cycle duration. It should be possible to gain some insights with pharmacological perturbations.

3) An interesting observation of this paper is that as the size of the tube is increased the number of waves starting at the boundary also increases. The authors propose that this corresponds to nuclei of higher intensity preferentially localizing close to the boundary, due to a geometrical effect causing higher competition among internal nuclei for import. The authors should present a clear analysis of the effects of nuclear density on the observed waves, as the nuclear-to-cytoplasmic ratio can influence cell cycle duration. Also, they do not report cell cycle durations and wave speeds in all the experiments with tubes of different geometry (I only found such data in the supplement for tubes of 100 micron). In the example shown in Figure 5, the cell cycle seems to lengthen in the small tube (as shown in the supplement) while it seems of constant duration in the large tube. Is that true? The authors should do more quantification of their data to provide arguments against alternative hypotheses. At this point, I remain a bit worried that there could be other explanations for the observed phenomena.

*Reviewer #2:*

Cyclin dependent kinase 1 (Cdk1) controls the entry into and exit from mitosis in eukaryotes. Previous studies from the Ferrell lab using frog egg extracts seeded with demembranated sperm (to serve as sources of DNA for nucleus assembly) demonstrated that Cdk1 activation proceeds as a bistable trigger wave through the extracts, providing an explanation for how the mitotic state is spatially coordinated. In the current study, Nolet et al. have used a similar approach to probe the relationship between nuclei, space and the organization of the Cdk1 trigger waves. They report that Cdk1 trigger waves often arise near particularly bright nuclei. From this they conclude that a) nuclei serve as pacemakers for trigger wave formation and b) that nuclei that are especially efficient at import are likely to serve as pacemakers. Both of these ideas are plausible, in that the nucleus is well-known to concentrate a variety of essential Cdk1 regulators. By a combination of modeling and experimental manipulation the authors further show that the spatial organization of the extract matters, with increased volume altering the distribution of pacemakers.

The findings of this study are important for at least two reasons. First, while nuclei are good candidates for pacemakers, their role in this capacity has yet to be empirically demonstrated (although the analysis of surface contraction waves in activated *Xenopus* eggs by Chang and Ferrell, 2013, makes a pretty strong case for nuclei as sites of Cdk1 wave initiation). Second, the relationship between pacemaker distribution and extract volume is a fascinating demonstration of the kind of nonintuitive but important emergent behaviors that can arise when dealing with bistable dynamics.

However, several points need to be addressed before the conclusions reached can be accepted. These are listed below:

The authors report "We noticed that the mitotic waves often originate close to nuclei that are considerably brighter than the surrounding nuclei (Figure 2A-C). We therefore hypothesized that a region with higher GFP-NLS intensity correlates with a higher local oscillation frequency, serving as a pacemaker that organizes the mitotic wave (Figure 2C). As a brighter nucleus is more efficient in the uptake of GFP-NLS, we reasoned that it similarly concentrates cell cycle regulators that lead to a local increase in the cell cycle frequency".

The study hinges upon the assumptions and conclusions implicit and explicit in these three sentences. It therefore follows that those assumptions and conclusions need to be well supported. The following suggestions are intended to help the authors provide such support.

"Often" and "brighter" are subjective terms. The definition of bright needs to be quantitatively established, as does the definition often, particularly as inspection of the data in Figure 2—figure supplement 5 provides as many clear examples of dim nuclei apparently acting as pacemakers (using the author's criteria) as bright nuclei acting as pacemakers, as well as examples of bright nuclei that do not apparently act as pacemakers. Ideally, the authors would provide a plot of nuclear GFP signal (normalized against a DNA marker such as DAPI) versus the number or frequency of trigger wave initiation associated with the nuclei.

Increased nuclear brightness may imply more efficient nuclear import but it may also imply greater age (and indeed, some of bright nuclei appear to have been around longer than their neighbors). It may also imply greater volume. That is, the kind of fluorescence imaging employed in these experiments is inherently nonlinear, and as a result, smaller nuclei may appear less bright simply because they have less GFP-NLS rather than less concentrated GFP-NLS. While one might imagine that all of the nuclei should be the same size, the standard preparation of demembranated sperm for egg extract studies often results in a population of samples that contains both intact and fragmented sperm. The authors could address their proposal that import efficiency is important by the application of pharmacological import inhibitors (along with a direct marker for DNA such as DAPI) which would, if their model is right, perturb the normal pattern or number of pacemakers. This would require a different marker for cell cycle progression, a point addressed below.

Ideally, testing the role of nuclei as pacemakers would entail a marker of mitotic entry that does not rely on the nuclei themselves. Three possibilities present themselves: The first is the FRET-based Cdk1 activity probe developed by the Pines lab. However, it may have insufficient spatial resolution for these studies. The second would be a microtubule probe, such as fluorescent tubulin or one of the commercially-available, fluorescent taxol derivatives (e.g. SiR-tubulin). Microtubule growth in M-phase extracts is limited but quite robust in interphase extracts. The third would be a probe for F-actin as it has been shown that M-phase extracts differ in their organization of F-actin than Interphase-arrested extracts (Field et al., 2011). This isn't to imply that the authors would need to repeat all of their experiments with a nucleus-independent cell cycle marker, but rather that they could simply run the basic assay in the presence of such a marker and determine if Cdk1 activation is more often correlated with the presence of a nucleus than would be predicted by chance. This point could be determined by comparing the number of trigger waves associated with nuclei in the original videos and then doing so again after rotating the channel showing the nuclei by 180 degrees. Additionally, the authors could compare the abundance and distribution of pacemakers in extracts with and without nuclei.

The authors note that others have proposed that the centriole/centrosome may serve as a pacemaker. A large fraction of demembranated *Xenopus* sperm usually have associated centrioles. The centrioles give rise to centrosomes in extracts and the centrosomes, in turn, generate microtubule asters. Because cyclin B binds to microtubules (via interactions with MAPs; see for example J. Cell Biol. 1995. 128:849-862) it is a distinct possibility that the microtubule organizing center generated by the demembranated sperm that account for the pacemaking activity. This point could be addressed by adding purified DNA to the extracts which will still assemble into nuclei (Newmeyer et al., 1986).

*Reviewer #3:*

The paper studies mitotic waves in cell-free *Xenopus* extracts, using a combination of experiments and modeling, analyzed primarily via numerical simulations. The main result is the finding that mitotic wave can be initiated either from the boundary or from "stronger" internal nuclei (within the model, having a higher frequency), and that which of the two wins is determined in part by the system's geometry. Some of the qualitative conclusions (e.g., that wider systems support wave initiation from the boundaries) are then verified experimentally.

The contents of the paper are carefully prepared and well-written, and the supporting information detailed and considerate. In particular, the modeling is explained well and implemented carefully. Nonetheless, the paper has some issues that should be addressed:

1) What is the biological significance – if any – whether the waves are boundary-driven or internally driven? The significance is given more strongly in the conclusion (starting in the third paragraph of the Discussion) than in the Introduction, and even then the results are discussed in the context of a different organism (*Drosophila*) than the one studied by the authors.

2) It would be good if the authors could put their results in the broader context of diffusion-reaction systems and, in particular, other biological systems such as the Min oscillations briefly mentioned in the Discussion – e.g., are the results expected to hold for other systems? This seems to be the implicit claim, since the features they note are conserved across two different models (the detailed biochemical model and the generic FHN model).

3) At times, it is hard to follow the logical flow of the paper – e.g., why do higher frequencies dominate? (This is mentioned only in the Discussion, without providing intuition). The choice of a higher frequency nuclei was given early in the paper without adequate explanation. Could the authors have instead used nuclei with deeper Gaussian wells? Or varying other parameters?

4) In the presentation of Figure 2, the authors show what appears to be transient experimental data (in panel B), then support this data with non-transient numerical simulation (in panels D and E) that shows the same spatiotemporal structure as the experiments. However, it is not clear to me why this comparison is valid, or why they assume only one prominent pacemaker region in panels D/E when there are multiple nuclei present and their assumption is that nuclear import drives pacemaking.

5) "While this simulation with a single nucleus closely relates to the *in vivo* situation of a typical cell" – where is this shown/discussed?

6) Figure 3 has several issues that require attention, most notably the presentation of the enhancement of *Cavg*. Firstly, The observed enhancement of *Cavg* near the boundary is much smaller than the values of 20-40% cited in Figure 2. Secondly, it is not clear whether the enhancement at the boundary is due to sequestration competition as the authors propose and model in this figure or the enhanced concentration resulting from the nearby boundary. When the nucleus is deconstructed after interphase, the proteins diffuse away; with a boundary nearby, the local concentration can be enhanced as it provides a barrier to diffusion. Is there a way to tease apart the two effects and identify which is dominant in the experiments? Lastly, the color scale in Figure 3D is misleading without a colorbar.

7) "Figure 3B shows that proteins quickly build up in the nuclear region in the early phase of the import period and then the proteins quickly disperse after nuclear envelope breakdown." This is unclear to me. The authors should maybe show a time trace of the concentration at x = 1.2mm, or some other, more interpretable plot.

8) Also, in the first paragraph of the subsection “Wider systems lead to boundary-driven mitotic waves”, results from the model are quoted, but it seems to me that they were not clearly presented beforehand? In particular, the authors show that nuclei near the boundary are more efficient at nuclear import, but never explicitly show that this results in boundary-driven waves. (This result is implied when Figure 3 is combined with Figure 2, but the paper lacks detailed end-to-end modeling.)

9) How are the kymograph lines determined in e.g. Figure 5? It is not clear if they are fit by hand, by computer, etc. and as a result, one is led to wonder how solid the conclusions from Figure 5 are. Additionally, the subtraction procedure in the blue-colored sub-panel of Figure 5A, B, C seems arbitrary.

10) There are several issues regarding the discussion of the boundary and/or dimensionality. First, the authors explore the effect of the boundaries at the ends of the tubes, but there is also a boundary azimuthally. Is there any way to determine whether this boundary plays a role in the initiation of the waves? Secondly, it is not clear what sets the length scale of the boundary-driven waves. Why is there a transition around 300 micron thick tubes? (This latter point may not be easy to answer from first principles.) Lastly (minor), around the fourth paragraph of the subsection “Wider systems lead to boundary-driven mitotic waves”, the authors discuss a two-dimensional system. They assert that the height of the droplet of extract relative to the diameter of the droplet determines the system dimensionality; this is not accurate; rather, the height of the droplet should be compared to the length scale of the wave (see a recent preprint for details: https://www.biorxiv.org/content/10.1101/2019.12.27.887273v1).

[Editors' note: further revisions were suggested prior to acceptance, as described below.]

Thank you for resubmitting your work entitled "Nuclei determine the spatial origin of mitotic waves" for further consideration by *eLife*. Your revised article has been reviewed by three peer reviewers, one of whom is a member of our Board of Reviewing Editors and the evaluation has been overseen by Naama Barkai as the Senior Editor.

All three reviewers agree that the paper is significantly improved by your new experiments, analyses and edits. However, there are few points that still need to be addressed before the paper can be accepted. One point (point 1) might require a bit more analysis or could be addressed by writing cautionary notes about the strength of conclusions that can be drawn from the uptake model. The remaining points (point 2-5) should all be addressed by textual modifications.

1) There are still some logical gaps in the paper which impact the readability and theory-to-experiment correspondence. For instance, there is still not a strong quantitative link between the nuclear import model presented in Figure 3 and the resulting wave dynamics explored elsewhere. Panels D and E in Figure 3 show that uptake competition is the main driver of concentration increases at nuclei near the boundaries of a given system, at least within their uptake model. What remains unclear is how robust their conclusions are – i.e., whether their conclusions hold for other plausible uptake models. Thus, it remains unclear how universal their uptake model conclusions are – in other words, whether the conclusions hold for other plausible modeling choices. It would be useful to show that either (a) quantitative evidence that their modeling choices provide testable predictions that are validated in experiment or (b) many model classes lead to the qualitative phenomena they observe. If these analyses cannot be performed the authors should at least comment on the fact that their results hold for a specific choice of the uptake model and that it remains unclear how generalizable the results are.

2) Some of the analysis is still somewhat opaque. For instance, there is not a clearly visible wave in Figure 2C. It is also not clear how the choices to draw (or not draw) wave fronts are made in the analysis leading to Figure 5 – which is a central point of the paper; for instance, in panel C of Figure 5—figure supplement 1, the wave fronts could be drawn in multiple ways. In response to our concerns, the authors have noted their image analysis methods include "drawing lines by visually judging the disappearance of nuclei" and that they "show the robustness of the findings by changing the fitting parameters used in the analysis over a wide range". It is still not clear whether these lines are drawn manually or in a regimented, computerized fashion. We trust that they have done this procedure carefully enough that their conclusions would not be affected by whatever handling method they have chosen, but it would be nice to have a clearer description of what has been done as well as a more substantive supplementary dataset, e.g., an expanded version of Figure 5—figure supplement 1.

3) The new data on the dependency of mitotic wave speed on cell cycle timing shown in Figure 2A are very interesting. The observed dependency is difficult to reconcile with trigger waves and rather suggests a transition from sweep to trigger waves as the cell cycle slows down, as suggested by Vergassola et al., 2018. That must be stated. For example, authors could say: "The dependency of mitotic wave speed on cell cycle duration is consistent with a transition from sweep waves to trigger waves as cell cycle slows down, as proposed in Vergassola et al., 2018".

4) The claims on the relevance of this paper findings to *Drosophila* must be further toned down. Even if the authors simulated sweep waves the internuclear distances used in the simulations are at least an order of magnitude higher than what is observe in *Drosophila*. So, in the fifth paragraph of the Discussion the authors should say something like: "However, the internuclear distance of our simulations is significantly larger than the one observed in the more crowded *Drosophila* embryo, so it remains unclear whether our results can directly extend to that system".

5) The importance of nuclear domains in *Drosophila* is an old concept named "energid". The authors should be clear about that in the second paragraph of the subsection “A computational model where nuclei spatially redistribute cell cycle regulators predicts the location of pacemaker regions”. It is also a bit misleading that the authors seem to imply that the nuclear domain of syncytial muscle or *Drosophila* embryos are similar in size to the ones observed in their experiments. They should comment on those importance size differences

*Reviewer #1:*

The manuscript by Nolet et al. is significantly improved by the revisions. I only have few minor requests of textual changes prior to recommending acceptance.

1) The dependency of the speed of waves with cell cycle timing is important. The data show that for very fast cell cycles one gets a wave speed that depends strongly on the cell cycle period and that for slow cycles the speed of the wave is more or less fixed. This is exactly the scenario proposed in the paper by Vergassola et al., 2018 with slow drives resulting in bistable waves and fast drives generating more rapid sweep wave with a clear dependence on the rate. That needs to be stated.

2) There is a significant difference in the internuclear distance observed in *Xenopus* extract and *Drosophila* embryos. I appreciate the effort of the authors to simulate sweep waves, but the internuclear distance in fly embryo is <~10 micron which is at least an order of magnitude lower of the values simulated by the authors. So, in their Discussion they should explain this by saying something like: However, the internuclear distance of our simulations is significantly larger than the one observed in the more crowded *Drosophila* embryo, so it remains unclear whether our results can directly extend to that system.

3) The importance of nuclear domains for nuclear migration was beautifully supported by data from Telley et al., 2012 in *Drosophila*, but it is an older concept and usually referred to as "energid". The authors should clarify that. It is also misleading that the authors seem to imply that the nuclear domain (including *Drosophila*) is similar to the one in their experiments. As said in point 2 the difference is about an order of magnitude and cannot be overlooked.

*Reviewer #2:*

The authors have carefully and thoroughly addressed the points I raised and, in every case, their original conclusions were confirmed.

*Reviewer #3:*

Overall, the modeling presentation is improved as there is less clutter after having moved some of the material to the SI. There are still, however, some logical gaps in the paper which impact the readability and theory-to-experiment correspondence. For instance, there is still not a strong quantitative link between the nuclear import model presented in Figure 3 and the resulting wave dynamics explored elsewhere. Additionally, it remains unclear how universal their uptake model conclusions are – in other words, whether the conclusions hold for other plausible modeling choices. It would be useful -though perhaps beyond the scope of this paper – to show that either (a) quantitative evidence that their modeling choices provide testable predictions that are validated in experiment or (b) many model classes lead to the qualitative phenomena they observe. Some additional points:

1) The authors have clarified their results in relation to our question about the relative impacts of boundary-condition-driven concentration increases versus competition-driven concentration increases. Panels D and E in Figure 3 show that uptake competition is the main driver of concentration increases at nuclei near the boundaries of a given system, at least within their uptake model. What remains unclear is how robust their conclusions are – i.e., whether their conclusions hold for other plausible uptake models.

2) Some of the analysis is still somewhat opaque. For instance, there is not a clearly visible wave in Figure 2C. It is also not clear how the choices to draw (or not draw) wave fronts are made in the analysis leading to Figure 5 – which is a central point of the paper; for instance, in panel C of Figure 5—figure supplement 1, the wave fronts could be drawn in multiple ways. In response to our concerns, the authors have noted their image analysis methods include "drawing lines by visually judging the disappearance of nuclei" and that they "show the robustness of the findings by changing the fitting parameters used in the analysis over a wide range". It is still not clear whether these lines are drawn manually or in a regimented, computerized fashion. We trust that they have done this procedure carefully enough that their conclusions would not be affected by whatever handling method they have chosen, but it would be nice to have a clearer description of what has been done as well as a more substantive supplementary dataset, e.g., an expanded version of Figure 5—figure supplement 1.

---

## [Author Response]

Essential revisions:1) Biological Significance. It is unclear what the biological significance of the presented results is. The proposal that the experiments will apply to *Drosophila* embryos is complicated by several observations in that system that argue for significant differences from your results: a) Cdk1 waves are not trigger waves in fly embryos (Vergassola et al., 2018); b) the correlation length of the field of Cdk1 activity (~100 microns) is significantly larger than internuclear distance (<10 micron) suggesting that nuclei are unlikely to act in isolation in that system; c) the local nuclear-to-cytoplasmic ratio is important for timing the cell cycle. Thus, we recommend that you propose other possible *in vivo* applications of your insights from *in vitro* experiments.2) Evidence for nuclei acting as pacemakers. The evidence that nuclei act as a pacemaker should be strengthened. There are several suggestions from the reviewers. While answering all the criticism might not be necessary, we would like you to address the main concerns of reviewer 2, as well as provide evidence that nuclear brightness is a better predictor of wave initiation than a measurement of the nuclear-to-cytoplasmic ratio (for example you could divide the extract in cell-like domains by some geometric tessellation and check whether nuclear brightness is a better predictor of wave initiation than the size of the cell-like domain). While FRAP experiments would be a great quantitative test of nuclear import/export parameters, they are not necessary if you lack the setup for performing them.3) Theoretical analysis. Reviewer 3 raises several theoretical questions that should be dealt with.Reviewer #1:Signaling waves are emerging as a general mechanism of regulation of biological processes. Among them, mitotic waves in early embryos are particularly interesting as they are both functionally important and a great system for quantitative dissection of the mechanisms of the waves. The paper by Nolet and collaborators addresses the mechanisms by which certain regions can emerge as pacemakers in large spatial systems. The authors use the *Xenopus* extract system which is able to undergo repeated cell cycles which organize in mitotic waves. The authors argue for two major conclusions: waves tend to start at nuclei that are more efficient at nuclear import; spatial dimensions influence whether waves tend to start near the boundary or in the middle of the domain. These are in principle interesting observations, but this paper suffers from significant limitations that need to be addressed. Specifically the significance of their *in vitro* findings for *in vivo* systems is unclear and the experimental analysis could be strengthen by more quantifications and additional experiments.1) While the experiments presented in this paper are well-executed, their relevance to *in vivo* systems remain unclear. The authors speculate that their results could explain the origin of waves in *Drosophila* embryos. However, this proposal is problematic and it is important that the authors address the fact that mitotic waves in *Drosophila* are not trigger waves (Vergassola et al., 2018). The authors need to study wave origin in the context of the mechanism proposed by Vergassola et al. if they wanted to make claims about the *Drosophila* embryo. Also notice that the inter-nuclear distances in these two systems are significantly different. Overall, I am very skeptical that one can generalize the findings of this paper to the fly embryo.

We agree that we could have done a better job on clarifying the *in vivo* relevance of our work. We have now expanded such a discussion both in the Introduction and in the Discussion by including (parts of) the Discussion below.

While several studies have addressed the potential biochemical mechanisms of mitotic waves (Chang and Ferrell Jr., 2013; Deneke et al., 2016; Vergassola et al., 2018), what determines the spatial origin of mitotic waves remains unclear. We find that mitotic waves originate at nuclei, which act as so-called pacemakers, regions that oscillate faster than their surroundings.

One advantage to having the nucleus control the timing of mitosis is that it allows the cell to ensure that DNA replication has completed before initiating mitosis. While DNA checkpoints are largely silenced in the early *Xenopus* embryo (Newport and Dasso, 1989, in *Drosophila* DNA content is known to activate the DNA-replication checkpoint and alter the cell cycle period (Farrell and O’Farrell, 2014; Deneke et al., 2016). A failure in the correct regulation of mitosis is associated with polyploidy, which plays a key role in nonmalignant physiological and pathological processes (Fox and Duronio,2013). In the absence of a proper pacemaker, or if the pacemaker were to be located elsewhere, linking DNA replication to mitosis would be more complicated and, perhaps, more prone to error.

Previous studies have pointed to the critical role of the nucleus in spatial redistributing cell cycle regulators (Gavet and Pines, 2010; Santos et al., 2012). In particular, the nuclear import of Cyclin B has been shown to lead to spatial positive feedback, ensuring a robust and irreversible mitotic entry (Santos et al., 2012). Nuclei have also been found to be crucial in ensuring cell cycle oscillations in the *Drosophila* embryo (Huang and Raff, 1999; Deneke et al., 2019). Interestingly, although previous reports have suggested that centrosomes serve as pacemakers (Chang and Ferrell Jr., 2013; Ishihara et al., 2014), we found that they are dispensable. After treating extracts with purified DNA, which lacks centrosomes, we still observed mitotic waves.

Nuclei are a natural choice of pacemaker for mitotic waves because they allow for a natural way to link one biological process, DNA replication, with another, mitosis. We hope that our work will further trigger new studies into the origin of pacemakers as the initiation of biological decisions mediated by traveling waves seem to be key in the proper coordination of a biological process. Traveling waves have, for example, also been found to propagate apoptosis, action potentials, and calcium signals over large distances. In these systems, defective mitochondria, signals from neighboring neurons, or fertilization serve as the initial trigger to locally activate a wave.

We also found that the interaction of multiple nuclei in a shared cytoplasm can lead to unexpected behavior, i.e. boundary-driven mitotic waves occur in wider systems. Strikingly, mitotic waves in the early *Drosophila* embryo have also been reported to often originate at the boundary (Foe and Alberts, 1983). During nuclear cycles 10-13 in the syncytial blastoderm of these early embryos, nuclei enter (and exit) mitosis in waves that originate metachronously from the opposite anterior and posterior poles of the embryo and terminate in its mid-region.

It was not our intention to make strong claims on the wave origin in the *Drosophila* embryo as we do not have experimental data to support such claims. However, we did feel that it was worthwhile to highlight that waves have been reported to often originate at the boundary in the *Drosophila* embryo. This observation is certainly intriguing as we are not aware of many biological examples where waves originate at the boundary. On the one hand, we have now formulated this discussion more carefully in the manuscript. On the other hand, while being cautious in our claims concerning the relevance of our findings for the *Drosophila* embryo, we have carried out additional simulations that support the idea that the existence of boundary-driven waves in *Drosophila* could be explained by the model presented in our manuscript. Here below we address key findings, parts of which we have also incorporated into the main text of the manuscript:

· *Vergassola* et al. provide evidence that mitotic waves in *Drosophila* are not trigger waves, but rather so-called sweep waves (Vergassola et al., 2018). However, in the same manuscript, the authors do show that slower and clearer trigger waves exist in mutant embryos (Vergassola et al., 2018). The experiment where mitotic waves in such mutant embryos are shown in this manuscript correspond to boundary-driven waves (see Figure 2 in Vergassola et al., 2018).

· Our work shows that boundary-driven waves can be generated for different types of oscillators, as long as their frequency is higher towards the boundary (Figure 4—figure supplements 1 and 2). This argues that the main ingredient for boundary-driven waves is the spatial redistribution process due to competing nuclei, which could also be relevant in *Drosophila*. The fact that Cdk1 oscillations only occur close to nuclei illustrates such a central role of nuclei (Deneke et al., 2019). Moreover, the uniform positioning of nuclei in the early *Drosophila* embryo (Deneke et al., 2019) is also a crucial element in our model to get boundary-driven waves. While this is definitely no proof that our work can be generalized to *Drosophila,* we do believe this is at least an interesting observation.

· Note that the work in (Vergassola et al., 2018) does not describe an oscillatory system. Instead, the model for sweep waves is based on a single propagating front in a bistable system where the locations of the saddle-node points (and the whole force field) changes in time (hence, everything is “sweeped”). We expanded this model by turning it into an oscillating system and assuming a higher cell cycle frequency towards the boundary. In doing so, we found that both trigger waves (with a fixed bistable switch or saddle-node points that are changed slowly) and sweep waves (saddle-node points are changed sufficiently fast) produced boundary-driven waves, but sweep waves were faster (see Figure 4—figure supplement 2). This is consistent with the work by *Vergassola* et al. and provides a potential mechanism to explain the spatial origin of sweep waves in *Drosophila*. In the model by *Vergassola* et al. the wave origin is purely random due to noise. The spatial redistribution process by a regular nuclear pattern in our work could be an additional effect to (perhaps partially) determine the wave origin.

· Finally, we checked the effect of the internuclear distance and the range of interaction/competition between nuclei. We find that boundary-driven waves still exist for larger interaction ranges (Figure 3—figure supplement 2) and smaller internuclear distances (Figure 3—figure supplement 4), a situation that more closely corresponds to the *Drosophila* system as mentioned by the reviewer.

2) The authors make the claim on more efficient nuclear import based on nuclear fluorescent intensity. However, it is my impression that many of the nuclei in the system do not separate following mitosis suggesting that these nuclei might ultimately become polyploid. So, I find nuclear fluorescence and intensity difficult to interpret. A better test of the nuclear import efficiency would be to use FRAP to measure the kinetic parameters of nuclear import and export. It would also be important to test how large the difference in nuclear import need to be to impact cell cycle durations. It is unclear how physiological nuclear dynamics is in the egg extract. Finally, it is in principle possible that the cell cycle influences nuclear import and vice versa. This would leave a bit of chicken/egg problem. Is nuclear import causing faster cell cycles or is higher nuclear import a consequence of the faster cell cycles? This paper presents no attempt at manipulating nuclear inputs or cell cycle duration. It should be possible to gain some insights with pharmacological perturbations.

The reviewer raises valid points. Several open questions related to how DNA and nuclei affect the spatiotemporal coordination of the cell cycle remain. We agree that it is important to attempt to manipulate nuclear inputs or cell cycle duration to verify some of our claims. To this end we supplemented cell-free extracts with different concentrations of Importazole, which is an inhibitor of importin-beta transport receptors. We also disrupted microtubule dynamics by adding *S*-Trityl-L-cysteine (STLC), a kinesin Eg5 inhibitor. Our experiments show that such perturbations influence local cell cycle duration and mitotic wave organization. Increasing inhibition of nuclear import leads to an average increase in the cell cycle period, smaller nuclei, and a loss of mitotic waves. This further confirms the crucial role that nuclear import processes play in forming proper nuclei and ensuring mitotic wave formation. We now discuss these results Figure 2E-G, Figure 2—figure supplement 1, and Figure 2—video 2.

The reviewer wonders whether many of the nuclei in the system do not separate following mitosis and whether the nuclear behavior in extracts is physiological or not. Various works of Levy and Heald have illustrated that nuclei expand *in vivo* at a rate comparable to that of egg frog extracts, illustrating that extracts faithfully recapitulate nuclear dynamics in the early embryo (Levy and Heald, Cell 2010). Cheng and Ferrell also recently observed that cell-like compartments in frog cycling extracts can undergo consecutive cell division cycles, where the daughter compartment contained a single nucleus each (Cheng and Ferrell, 2019). In our case, such divisions are hard to see with our GFP-NLS reporter, likely due to the fact that we study extracts contained in tubes rather than thin sheets, such that divisions are likely to occur outside the plane of the microscope. In order to explore physiological concentrations of nuclear densities in the egg extract, we explored the behavior for four different nuclear densities (approx. 30, 60, 250 nuclei/μl, and no nuclei). Figure 2A-C and Video 1 show the results of these experiments, where we found that the cell cycle period decreases and the wave speed increases as the nuclear density increases. These experiments show that in the absence of nuclei, the system still oscillates, but is unable to form clear mitotic waves. This directly illustrates the essential role that nuclei play in generating mitotic waves.

The reviewer also touches on another interesting question: while the cell cycle controls nuclear import, nuclear import can also drive the cell cycle, which leads to a chicken/egg problem. Is nuclear import causing faster cell cycles or is higher nuclear import a consequence of the faster cell cycles. In order to address the complex interaction between these processes we have developed a new theoretical model which expands the previous one. Our original approach consisted of two different models:

a) Model 1 (corresponding to previous Figure 3) described the spatial redistribution of cell cycle regulators by competing nuclei including import and diffusion processes that were timed by a constant cell cycle period.

b) In Model 2 (corresponding to previous Figure 4), the spatial profile of cell cycle period determinants obtained from Model 1 were introduced as a spatial profile in the cell cycle frequency in established partial differential equations for cell cycle oscillations, showing mitotic waves.

Now, however, we have expanded Model 1 by including a modulation of the cell cycle period based on the local density of cell cycle regulators. This integrated model directly explains the experimentally observed phenomena, such as boundary-driven waves and competing mitotic waves. We now include such detailed end-to-end modeling in the main text. We decided to move Model 2 to Figure 4—figure supplement 1, where it is used to illustrate the generic aspects of competition between waves generated by spatially heterogeneous frequency profiles.

3) An interesting observation of this paper is that as the size of the tube is increased the number of waves starting at the boundary also increases. The authors propose that this corresponds to nuclei of higher intensity preferentially localizing close to the boundary, due to a geometrical effect causing higher competition among internal nuclei for import. The authors should present a clear analysis of the effects of nuclear density on the observed waves, as the nuclear-to-cytoplasmic ratio can influence cell cycle duration. Also, they do not report cell cycle durations and wave speeds in all the experiments with tubes of different geometry (I only found such data in the supplement for tubes of 100 micron). In the example shown in Figure 5, the cell cycle seems to lengthen in the small tube (as shown in the supplement) while it seems of constant duration in the large tube. Is that true? The authors should do more quantification of their data to provide arguments against alternative hypotheses. At this point, I remain a bit worried that there could be other explanations for the observed phenomena.

In Figure 1B, Figure 1—figure supplement 1 and Figure 2A-C we now report in more detail on cell cycle periods and wave speeds for our experiments in thin tubes (100 μm, 200 μm) for different nuclear densities (approx. 30, 60, 250 nuclei/μl, and no nuclei). We found that the average cell cycle period increases over time (Figure 2B). Such increase in cell cycle period has been linked to a decrease in ATP supply over time (Guan et al., 2018). One additional explanation could be that a decrease in cell cycle period is related to increasing levels of DNA as it is replicated (Dasso and Newport, 1990). We found that extracts with less added sperm nuclei had a faster cell cycle, which is again consistent with the idea that increasing levels of DNA slow down the cell cycle (Figure 2A) (Dasso and Newport, 1990). Mitotic waves were similarly observed, but the wave speeds were initially faster than in tubes with a higher nuclear density (Figure 2B). The waves then slowed down to similar speeds as in the case with the higher concentration of sperm nuclei.

Interestingly, a decrease in nuclear density did not lead to a big change the internuclear distance (Figure 1—figure supplement 1I). Instead, it created more and larger regions where nuclei were absent (Figure 1—figure supplement 2), and pacemakers were predominantly found close to these regions (Figure 1—figure supplement 2). Cheng and Ferrell observed a similar transition from a regular pattern of equidistantly spaced nuclei to a system with holes in *Xenopus* interphase egg extracts when decreasing the concentration of added sperm nuclei (Cheng and Ferrell, 2019).

Next, we further decreased the nuclear density (approx. 30 nuclei/µl), such that only few nuclei remained in an entire tube. Here, we used a fluorescent microtubule reporter to visualize the spatial coordination of mitotic entry, while bright-field images were used to track the location of nuclei (Video 1). Mitotic waves were found to originate at the few nuclei present in the tube, and they traveled through the whole tube (several mm) at a speed of approx. 60 µm/min (Video 1, Figure 2C). In the absence of any nuclei in the tube (no added demembranated sperm nuclei), we still observed cell cycle oscillations, but no mitotic waves were observed (Video 1, Figure 2B). These experiments underscore the critical role that nuclei play in changing the cell cycle period and introducing sufficient spatial heterogeneity in cell cycle period to generate clear mitotic waves.

In Figure 5—figure supplement 5 we also characterized the wave speed and cell cycle duration for thicker tubes (300 μm, 560 μm). This analysis shows that the cell cycle period also increases in time in thicker tubes, but this increase is indeed slower as observed by the reviewer. The cell cycle period is also found to be initially longer for thicker tubes. Moreover, the wave speeds are found to be generally lower for thicker tubes.

Finally, we have extended our analysis to see whether GFP-NLS intensity is a good predictor of local cell cycle period and pacemaker location. We analyzed the spatial GFP-NLS intensity profile, the spatial profile of cell cycle periods, and the internuclear distance (Figure 1B). As a brighter nucleus has taken up more GFP-NLS, we reasoned that it similarly concentrates cell cycle regulators that lead to a local increase in the cell cycle frequency. We directly correlated this with the local period, which indeed showed that this region oscillated faster (Figure 1B). To further understand why certain nuclei were brighter, we explored whether their environment had any particular characteristics. We characterized the distance between the different nuclei and found that they were typically separated by 150 − 200µm(Figure 1—figure supplement 1). However, we found that the brightest nucleus is also most separated from its neighboring nuclei (Figure 1B). This finding is consistent with the idea that nuclei increase their oscillation frequency by concentrating cell cycle regulators, as they have a larger pool of regulators in their surroundings to import. We analyzed the spatial GFP-NLS intensity profile and the internuclear distance for 9 other experiments where we could clearly identify nuclei and mitotic waves. Overall, in 90% of the analyzed experiments the pacemaker location was well predicted by the region with the highest GFP-NLS intensity and/or the region where nuclei were most separated from their neighboring nuclei (Figure 1A, B and Figure 1—figure supplements 2 and 3). The total nuclear GFP-NLS intensity was also found to be a better indicator of the pacemaker location than the nuclear size as indicated by Hoechst staining, or than the GFP-NLS intensity normalized to the Hoechst signal (Figure 1—figure supplement 3).

Reviewer #2:Several points need to be addressed before the conclusions reached can be accepted. These are listed below:The authors report "We noticed that the mitotic waves often originate close to nuclei that are considerably brighter than the surrounding nuclei (Figure 2A-C). We therefore hypothesized that a region with higher GFP-NLS intensity correlates with a higher local oscillation frequency, serving as a pacemaker that organizes the mitotic wave (Figure 2C). As a brighter nucleus is more efficient in the uptake of GFP-NLS, we reasoned that it similarly concentrates cell cycle regulators that lead to a local increase in the cell cycle frequency".The study hinges upon the assumptions and conclusions implicit and explicit in these three sentences. It therefore follows that those assumptions and conclusions need to be well supported. The following suggestions are intended to help the authors provide such support."Often" and "brighter" are subjective terms. The definition of bright needs to be quantitatively established, as does the definition often, particularly as inspection of the data in Figure-2—figure supplement 5 provides as many clear examples of dim nuclei apparently acting as pacemakers (using the author's criteria) as bright nuclei acting as pacemakers, as well as examples of bright nuclei that do not apparently act as pacemakers. Ideally, the authors would provide a plot of nuclear GFP signal (normalized against a DNA marker such as DAPI) versus the number or frequency of trigger wave initiation associated with the nuclei.Increased nuclear brightness may imply more efficient nuclear import but it may also imply greater age (and indeed, some of bright nuclei appear to have been around longer than their neighbors). It may also imply greater volume. That is, the kind of fluorescence imaging employed in these experiments is inherently nonlinear, and as a result, smaller nuclei may appear less bright simply because they have less GFP-NLS rather than less concentrated GFP-NLS.

We have extended our analysis to see whether GFP-NLS intensity is a good predictor of local cell cycle period and pacemaker location. We analyzed the spatial GFP-NLS intensity profile, the spatial profile of cell cycle periods, and the internuclear distance (Figure 1B). As a brighter nucleus has taken up more GFP-NLS, we reasoned that it similarly concentrates cell cycle regulators that lead to a local increase in the cell cycle frequency. We therefore directly correlated this with the local period, which indeed showed that this region oscillated faster (Figure 1B). To further understand why certain nuclei were brighter, we explored whether their environment had any particular characteristics. We characterized the distance between the different nuclei and found that they were typically separated by 150 − 200µm (Figure 1—figure supplement 1). However, we found that the brightest nucleus is also most separated from its neighboring nuclei (Figure 1B). This finding is consistent with the idea that nuclei increase their oscillation frequency by concentrating cell cycle regulators, as they have a larger pool of regulators in their surroundings to import. We analyzed the spatial GFP-NLS intensity profile and the internuclear distance for 9 other experiments where we could clearly identify nuclei and mitotic waves. Overall, in 90% of the analyzed experiments the pacemaker location was well predicted by the region with the highest GFP-NLS intensity and/or the region where nuclei were most separated from their neighboring nuclei (Figure 1A, B and Figure 1—figure supplements 2 and 3).

We then repeated several experiments both with the GFP-NLS reporter and a Hoechst marker for DNA. In Figure 1—figure supplement 3 we show the analysis of an experiment with a clear mitotic wave. We first repeated the analysis of the spatial distribution of the total and maximal GFP-NLS intensity, as well as the internuclear distance. Again, both measures peaked close to the pacemaker location. Next, we analyzed the spatial distribution of the Hoechst signal to determine the internuclear distance and the nuclear size. While the internuclear distance was again high close to the pacemaker location, the nuclear size was not clearly higher at the pacemaker location. We then used the measure of nuclear size to renormalize the total and maximal GFP-NLS signal. In such normalization the measure was no longer larger than its surroundings close to the pacemaker location. This argues that the higher GFP-NLS intensity associated to pacemakers is due their larger size and volume and not due to having more concentrated GFP-NLS. We therefore no longer mention nuclear import efficiency in the manuscript, but have adopted these new measures to discuss pacemaker location. We thank the reviewer for pointing out the important difference between having a larger GFP-NLS signal due to larger total nuclear import or due to a higher nuclear import efficiency.

While one might imagine that all of the nuclei should be the same size, the standard preparation of demembranated sperm for egg extract studies often results in a population of samples that contains both intact and fragmented sperm. The authors could address their proposal that import efficiency is important by the application of pharmacological import inhibitors (along with a direct marker for DNA such as DAPI) which would, if their model is right, perturb the normal pattern or number of pacemakers. This would require a different marker for cell cycle progression, a point addressed below.

We agree that it is important to attempt to manipulate nuclear inputs or cell cycle duration to verify some of our claims. To this end we supplemented cell-free extracts with different concentrations of Importazole, which is an inhibitor of importin-beta transport receptors. We also disrupted microtubule dynamics by adding *S*-Trityl-L-cysteine (STLC), a kinesin Eg5 inhibitor. Our experiments show that such perturbations influence local cell cycle duration and mitotic wave organization. Increasing inhibition of nuclear import leads to an average increase in the cell cycle period, smaller nuclei, and a loss of mitotic waves. This further confirms the crucial role that nuclear import processes play in forming proper nuclei and ensuring mitotic wave formation. We now discuss these results (see Figure 2E-G, Figure 2—figure supplement 1, and Figure 2—video 2).

Ideally, testing the role of nuclei as pacemakers would entail a marker of mitotic entry that does not rely on the nuclei themselves. Three possibilities present themselves: The first is the FRET-based Cdk1 activity probe developed by the Pines lab. However, it may have insufficient spatial resolution for these studies. The second would be a microtubule probe, such as fluorescent tubulin or one of the commercially-available, fluorescent taxol derivatives (e.g. SiR-tubulin). Microtubule growth in M-phase extracts is limited but quite robust in interphase extracts. The third would be a probe for F-actin as it has been shown that M-phase extracts differ in their organization of F-actin than Interphase-arrested extracts (Field et al., 2011). This isn't to imply that the authors would need to repeat all of their experiments with a nucleus-independent cell cycle marker, but rather that they could simply run the basic assay in the presence of such a marker and determine if Cdk1 activation is more often correlated with the presence of a nucleus than would be predicted by chance. This point could be determined by comparing the number of trigger waves associated with nuclei in the original videos and then doing so again after rotating the channel showing the nuclei by 180 degrees. Additionally, the authors could compare the abundance and distribution of pacemakers in extracts with and without nuclei.

Thanks for the good suggestion to use an alternative marker for mitotic entry that does not rely on the nuclei themselves, which would allow perturbing the nuclear density and still have a way of visualizing mitotic wave dynamics. We repeated experiments with a microtubule reporter, using fluorescently labeled tubulin (HiLyte Fluor488). Figure 1C and Figure 1—video 2show that mitotic waves are also observed using such a microtubule reporter, as well as in bright-field. Using these alternative reporters, we then tested how critical system parameters such as nuclear density influences the mitotic wave dynamics.

In Figure 1B, Figure 1—figure supplement 1, and Figure 2A-C we analyzed cell cycle periods and wave speeds in additional experiments in thin tubes (100 μm, 200 μm) for different nuclear densities (approx. 30, 60, 250 nuclei/μl, and no nuclei). We found that the average cell cycle period increases over time (Figure 2B). Such increase in cell cycle period has been linked to a decrease in ATP supply over time (Guan et al., 2018). One additional explanation could be that a decrease in cell cycle period is related to increasing levels of DNA as it is replicated (Dasso and Newport, 1990). We found that extracts with less added sperm nuclei had a faster cell cycle, which is again consistent with the idea that increasing levels of DNA slow down the cell cycle (Figure 2A) (Dasso and Newport, 1990). Mitotic waves were similarly observed, but the wave speeds were initially faster than in tubes with a higher nuclear density (Figure 2B). The waves then slowed down to similar speeds as in the case with the higher concentration of sperm nuclei.

Interestingly, a decrease in nuclear density did not lead to a big change the internuclear distance (Figure 1—figure supplement 1I). Instead, it created more and larger regions where nuclei were absent (Figure 1—figure supplement 2), and pacemakers were predominantly found close to these regions (Figure 1—figure supplement 2). Cheng and Ferrell observed a similar transition from a regular pattern of equidistantly spaced nuclei to a system with holes in *Xenopus* interphase egg extracts when decreasing the concentration of added sperm nuclei (Cheng and Ferrell, 2019).

Next, we further decreased the nuclear density (approx. 30 nuclei/µl), such that only few nuclei remained in an entire tube. Here, we used the fluorescent microtubule reporter to visualize the spatial coordination of mitotic entry, while bright-field images were used to track the location of nuclei (Video 1). Mitotic waves were found to originate at the few nuclei present in the tube, and they traveled through the whole tube (several mm) at a speed of approx. 60 µm/min (Video 1, Figure 2C). In the absence of any nuclei in the tube (no added demembranated sperm nuclei), we still observed cell cycle oscillations, but no mitotic waves were observed (Video 1, Figure 2B). These experiments underscore the critical role that nuclei play in changing the cell cycle period and introducing sufficient spatial heterogeneity in cell cycle period to generate clear mitotic waves.

The authors note that others have proposed that the centriole/centrosome may serve as a pacemaker. A large fraction of demembranated *Xenopus* sperm usually have associated centrioles. The centrioles give rise to centrosomes in extracts and the centrosomes, in turn, generate microtubule asters. Because cyclin B binds to microtubules (via interactions with MAPs; see for example J. Cell Biol. 1995. 128:849-862) it is a distinct possibility that the microtubule organizing center generated by the demembranated sperm that account for the pacemaking activity. This point could be addressed by adding purified DNA to the extracts which will still assemble into nuclei (Newmeyer et al., 1986).

Centrosomes have indeed also been suggested to serve as pacemakers (Chang and Ferrell Jr., 2013; Ishihara et al., 2014), potentially by concentrating pro-mitotic factors such as Cdc25 and cyclin B (Bonnet et al., 2008; Jackman et al., 2003). As demembranated sperm nuclei also provide centrosomes that can generate microtubule asters, it is certainly relevant to test whether such centrosomes are critical to generate pacemakers. We therefore added purified DNA to the extracts, which assembled into nuclei (Newmeyer et al., 1986). Mitotic waves were still observed indicating that DNA alone is sufficient to create pacemaker-generated mitotic waves without a need for centrosomes (Figure 2D, Figure 1—video 2).

Reviewer #3:[…] The contents of the paper are carefully prepared and well-written, and the supporting information detailed and considerate. In particular, the modeling is explained well and implemented carefully. Nonetheless, the paper has some issues that should be addressed:1) What is the biological significance – if any – whether the waves are boundary-driven or internally driven? The significance is given more strongly in the conclusion (starting in the third paragraph of the Discussion) than in the Introduction, and even then the results are discussed in the context of a different organism (*Drosophila*) than the one studied by the authors.

We agree that we could have done a better job on clarifying the *in vivo* relevance of our work. We have now expanded such a discussion both in the Introduction and in the Discussion of the manuscript by including (parts of) our response below.

While several studies have addressed the potential biochemical mechanisms of mitotic waves (Chang and Ferrell Jr., 2013; Deneke et al., 2016; Vergassola et al., 2018), what determines the spatial origin of mitotic waves remains unclear. We find that mitotic waves originate at nuclei, which act as so-called pacemakers, regions that oscillate faster than their surroundings.

One advantage to having the nucleus control the timing of mitosis is that it allows the cell to ensure that DNA replication has completed before initiating mitosis. While DNA checkpoints are largely silenced in the early *Xenopus* embryo (Newport and Dasso, 1989), in *Drosophila* DNA content is known to activate the DNA-replication checkpoint and alter the cell cycle period (Deneke et al., 2016). A failure in the correct regulation of mitosis is associated with polyploidy, which plays a key role in nonmalignant physiological and pathological processes (Fox and Duronio, 2013). In the absence of a proper pacemaker, or if the pacemaker were to be located elsewhere, linking DNA replication to mitosis would be more complicated and, perhaps, more prone to error.

Previous studies have pointed to the critical role of the nucleus in spatial redistributing cell cycle regulators (Gavet and Pines, 2010; Santos et al., 2012). In particular, the nuclear import of Cyclin B has been shown to lead to spatial positive feedback, ensuring a robust and irreversible mitotic entry (Santos et al., 2012). Nuclei have also been found to be crucial in ensuring cell cycle oscillations in the *Drosophila* embryo (Huang and Raff, 1999; Deneke et al., 2019). Interestingly, although previous reports have suggested that centrosomes serve as pacemakers (Chang and Ferrell Jr., 2013; Ishihara et al., 2014), we found that they are dispensable. After treating extracts with purified DNA, which lacks centrosomes, we still observed mitotic waves.

Nuclei are a natural choice of pacemaker for mitotic waves because they allow for a natural way to link one biological process, DNA replication, with another, mitosis. We hope that our work will further trigger new studies into the origin of pacemakers as the initiation of biological decisions mediated by traveling waves seem to be key in the proper coordination of a biological process. Traveling waves have, for example, also been found to propagate apoptosis, action potentials, and calcium signals over large distances. In these systems, defective mitochondria, signals from neighboring neurons, or fertilization serve as the initial trigger to locally activate a wave.

We also found that the interaction of multiple nuclei in a shared cytoplasm can lead to unexpected behavior, i.e. boundary-driven mitotic waves occur in wider systems. Strikingly, mitotic waves in the early *Drosophila* embryo have also been reported to often originate at the boundary (Foe and Alberts, 1983). During nuclear cycles 10-13 in the syncytial blastoderm of these early embryos, nuclei enter (and exit) mitosis in waves that originate metachronously from the opposite anterior and posterior poles of the embryo and terminate in its mid-region. In this revised version of the manuscript, we have included additional simulations that support the idea that the existence of boundary-driven waves in *Drosophila* could be explained by the model presented in our manuscript.

2) It would be good if the authors could put their results in the broader context of diffusion-reaction systems and, in particular, other biological systems such as the Min oscillations briefly mentioned in the Discussion – e.g., are the results expected to hold for other systems? This seems to be the implicit claim, since the features they note are conserved across two different models (the detailed biochemical model and the generic FHN model).

In this revised version, we have tried to find a better balance between experiments and theory. On the one hand, we have reduced the parts in the main text where we discuss cell cycle oscillation models and the generic FHN model. This has allowed us to expand the experimental analysis, data analyses and biological interpretations. Instead, we have largely moved these modeling parts to supplementary figures. On the other hand, we have also expanded our modeling efforts by including more detailed end-to-end modeling, see point (8). As requested, we have also tried to put our results in a broader context of diffusion-reaction systems while trying to avoid making the manuscript too theoretical. We included (parts of) the discussions below in the manuscript.

We wondered whether these dynamics of competing pacemakers are specific to this particular computational model that includes nuclear import and diffusion processes. Therefore, we also implemented known PDE models of cell cycle oscillations (Appendix 2), where we define two pacemaker regions (see Figure 4—figure supplement 1A-C): an internal pacemaker and a boundary pacemaker region. We carried out simulations continuously changing the relative strength of both pacemaker regions by increasing the difference in cell cycle period. We found a gradual transition from boundary-driven dynamics to internal pacemaker-driven dynamics (Figure 4—figure supplement 1D). Similar results were found by using the FitzHugh-Nagumo oscillator model, a general model for relaxation-type oscillatory systems (Figure 4—figure supplement 1E). Oscillations are referred to as being of the relaxation-type when they are characterized by two separate timescales. One timescale is slow, i.e. cyclin abundances slowly build up in interphase, while the other is a fast activation of Cdk1.

We asked ourselves whether boundary-driven waves were specific to oscillatory systems based on relaxation-type oscillations. Therefore, we gradually decreased the timescale separation in the FitzHugh-Nagumo oscillator model, which led to more sinusoidal oscillations (Figure 4—figure supplement 2A) and preserved boundary-driven waves (Figure 4—figure supplement 2B). This suggests the generation of boundary-driven waves is largely independent of the type of oscillations, as long as the oscillation period is decreased close to the boundary. Our findings underscore the generic character of the dynamics of multiple competing pacemakers. Pacemaker-driven traveling waves, also often referred to as target patterns, have been widely studied and they form thanks to spatial heterogeneities that locally increase the oscillation frequency. The majority of such pacemaker waves were initially observed in chemical reaction-diffusion systems where heterogeneities were introduced as dust particles that locally modified the properties of the medium (Zaikin and Zhabotinsky, 1970; Zhabotinsky and Zaikin, 1973; Tyson and Fife, 1980). These experimental observations triggered many other studies on both traveling waves (Tyson and Fife, 1980; Kopell, 1981; Hagan, 1981; Kuramoto, 1984; Jakubith et al., 1990; Bugrim et al., 1996; Bub et al., 2005; Stich and Mikhailov, 2006) and spiral waves (Jakubith et al., 1990; Bub et al., 2002, 2005) triggered by a pacemaker. The interaction of multiple pacemaker waves has also been analyzed (Kuramoto, 1984; Walgraef et al., 1983; Mikhailov and Engel, 1986; Lee et al., 1996; Kheawon et al., 2007). In general, they propagate into the surrounding medium and compete with each other until the pacemaker with the highest frequency ultimately entrains the whole system (Kuramoto, 1984). The existence of the transition region is therefore somewhat surprising. However, simulating the system for increasingly longer transient times, we find that the transition region where boundary-driven waves and internal pacemaker-driven waves coexist shrinks, suggesting that after infinitely long transients one pacemaker indeed controls the whole domain. Such infinite transient times are, however, less biologically relevant as the early embryonic cell cycle oscillations only persist for about 13 cycles (Box 2). Therefore, one would expect to observe the full range of transient pacemaker dynamics in actual biological systems.

Our findings illustrate that the spatial environment has a strong influence on how biological processes self-organize. In particular, increasing the spatial dimensions of the system leads to a higher probability of observing mitotic waves that originate at the boundary of the system. Other studies have also stressed the importance of system size, boundaries, and geometry on self-organization processes. For example, using cell-free frog extracts, cytoplasmic volume was demonstrated to determine the spindle size (Good et al., 2013) and the size of the nucleus (Hara and Merten, 2015). System boundaries (Kopell et al., 1991; Haim et al., 1996; Rabinovitch et al., 2001; McNamara et al., 2016; Bernitt et al., 2017) and system geometry (Wettman et al., 2018) have been shown to affect the dynamics of traveling waves. In the widely studied amoeba *Dictyostelium discoideum*, the origin of cAMP waves has been studied in inhomogeneous systems. Waves appear spontaneously in areas of higher cell density with the oscillation frequency of these centers depending on their density (Vidal-Henriquez and Gholami, 2019). In the presence of advection, a boundary-induced instability was found to periodically excite a cAMP wave near the boundary (Vidal-Henriquez et al., 2017).

Another well-characterized model organism is the bacterium *Escherichia coli*, where Min-protein wave patterns help select the site of cell division (Hu and Lutkenhaus, 1999; Raskin and De Boer, 1999). Wave patterns and the location of cell division have been shown to strongly depend on the system size and geometry, both *in vivo* by deforming cell shape (Männik et al., 2013; Wu et al., 2015; Wettman et al., 2018) and *in vitro* by reconstituting Min oscillations in open and enclosed compartments (Zieske and Schwille, 2014; Zieske et al., 2016; Caspi and Dekker, 2016; Wettman et al., 2018). As thin compartments were gradually increased in length, multiple regions of oscillations were observed (Zieske and Schwille, 2014; Zieske et al., 2016; Caspi and Dekker, 2016; Wettman et al., 2018). For more complex geometries, many more wave patterns have been observed, such as standing waves, traveling planar and spiral waves, and coexisting stable stationary distributions (Zieske and Schwille, 2014; Zieske et al., 2016; Caspi and Dekker, 2016; Wettman et al., 2018). While there are similarities with our findings in the *Xenopus* cell-free extracts, one important difference is that the wave patterns in the Min system are mainly controlled by the spatial dimensions and geometry. In contrast, in our findings the influence of the spatial dimensions are, at least partially, mediated by the nuclei within the oscillatory medium that serve as pacemakers.

3) At times, it is hard to follow the logical flow of the paper – e.g., why do higher frequencies dominate? (This is mentioned only in the Discussion, without providing intuition). The choice of a higher frequency nuclei was given early in the paper without adequate explanation. Could the authors have instead used nuclei with deeper Gaussian wells? Or varying other parameters?

We should indeed have included a better explanation of a pacemaker at the start of the manuscript. We now discuss the link between pacemakers and oscillation frequency already in the Abstract, the Introduction, and the first section of the paper (subsection “Nuclei serve as pacemakers to organize mitotic waves”. This is discussed as follows:

Introduction: “We find that mitotic waves originate at nuclei, which act as so-called pacemakers, regions that oscillate faster than their surroundings (Kuramoto, 1984). […] We postulate that nuclei can concentrate cell cycle regulators, thus leading to faster cell cycle oscillations at those nuclear locations.”

Subsection “Nuclei serve as pacemakers to organize mitotic waves”: “We noticed that the mitotic wave originated close to a nucleus that is considerably brighter than the surrounding nuclei (Figure 1A). […] We directly correlated this with the local period, which indeed showed that this region oscillated faster (Figure 1B).”

As mentioned, we now no longer include simulations of cell cycle oscillations at the beginning of the manuscript as we feel the mix of modeling and experiment at the very start provided less of a logical flow (see also point (8)). However, as a short answer to the reviewer’s question: a lot of parameters can be varied to change the cell cycle period in the different models. We found that this had little effect and what matters most for all the dynamical behavior reported here is the fact that there is a clear frequency difference spatially. See e.g. also point (2) where we discuss the generality for different types of oscillators.

4) In the presentation of Figure 2, the authors show what appears to be transient experimental data (in panel B), then support this data with non-transient numerical simulation (in panels D and E) that shows the same spatiotemporal structure as the experiments. However, it is not clear to me why this comparison is valid, or why they assume only one prominent pacemaker region in panels D/E when there are multiple nuclei present and their assumption is that nuclear import drives pacemaking.

It is correct that the experiments are always in transient. Mitotic waves need time to build up in the presence of pacemaker regions. In fact, as we now report more elaborately in Figure 2, the cell cycle periods are also continuously changing in time. In the original modeling panels in the old Figure 2, we indeed showed dynamics that had already converged to well-defined waves (so non-transient behavior) and we did not implement a cell cycle period that was continuously changing. We believe that these assumptions make sense as they simplify the interpretation and modeling, while still capturing the essential dynamics. We both assumed one prominent pacemaker and multiple pacemakers associated to the multiple nuclei in the system, and we showed those two approaches to be valid in the old Figure 2 and old Figure 2—figure supplement 3. However, as mentioned before, partially motivated by point (8) by the reviewer, we chose to remove all these modeling figures at the beginning of the manuscript in favor of more detailed end-to-end modeling in new Figure 3 and 4.

5) "While this simulation with a single nucleus closely relates to the *in vivo* situation of a typical cell" – where is this shown/discussed?

We now rephrased this as follows:

While a typical cell contains a single nucleus, the cell-free extract experiment shown in *Figure 1* consists of many distributed nuclei.

6) Figure 3 has several issues that require attention, most notably the presentation of the enhancement of C_avg_. Firstly, The observed enhancement of C_avg_ near the boundary is much smaller than the values of 20-40% cited in Figure 2. Secondly, it is not clear whether the enhancement at the boundary is due to sequestration competition – as the authors propose and model in this figure or the enhanced concentration resulting from the nearby boundary. When the nucleus is deconstructed after interphase, the proteins diffuse away; with a boundary nearby, the local concentration can be enhanced as it provides a barrier to diffusion. Is there a way to tease apart the two effects and identify which is dominant in the experiments? Lastly, the color scale in Figure 3D is misleading without a colorbar.

We have added colorbars everywhere to clarify the scale of the changes in *C_avg_*. These changes are indeed much smaller that the frequency difference that we implemented in the model in the old Figure 2. This is because *C_avg_* represents a concentration of a certain cell cycle regulator. The actual cell cycle frequency is likely determined in a complicated (and currently largely unknown) way by a wide range of such regulators. It is possible that “small” relative changes in the concentration of cell cycle regulators can have a large effect on the cell cycle frequency.

It is indeed an interesting question whether the build-up of regulators at the boundary is due to either a sequestration effect, a boundary effect, or a combination of both. We have now carried out more simulations to tease them apart. We have discussed this in the following paragraph:

“The build-up of regulators at the boundary is mainly attributed to the fact that nuclei in the interior of the domain compete with neighboring nuclei to attract the available proteins, while nuclei close to the boundary only have one such “competitor". […] Instead, proteins build up close to the nuclei adjacent to the gaps (Figure 3—figure supplement 3).”

7) "Figure 3B shows that proteins quickly build up in the nuclear region in the early phase of the import period and then the proteins quickly disperse after nuclear envelope breakdown." This is unclear to me. The authors should maybe show a time trace of the concentration at x = 1.2mm, or some other, more interpretable plot.

We have now included such a time trace to clarify this more.

8) Also, in the first paragraph of the subsection “Wider systems lead to boundary-driven mitotic waves”, results from the model are quoted, but it seems to me that they were not clearly presented beforehand? In particular, the authors show that nuclei near the boundary are more efficient at nuclear import, but never explicitly show that this results in boundary-driven waves. (This result is implied when Figure 3 is combined with Figure 2, but the paper lacks detailed end-to-end modeling.)

This is a very good point. In order to improve the logical flow and clarity of the results, we have developed a new theoretical model which expands the previous one. Our original approach consisted of two different models:

a) Model 1 (corresponding to Figure 3) described the spatial redistribution of cell cycle regulators by competing nuclei including import and diffusion processes that were timed by a constant cell cycle period.

b) In Model 2 (corresponding to Figure 4), the spatial profile of cell cycle period determinants obtained from Model 1 were introduced as a spatial profile in the cell cycle frequency in established partial differential equations for cell cycle oscillations, showing mitotic waves.

Now, however, we expanded Model 1 by including a modulation of the cell cycle frequency based on the local density of cell cycle regulators. This integrated model directly explains the experimentally observed phenomena, such as boundary-driven waves and competing mitotic waves. We now include such detailed end-to-end modeling in the main text. We decided to move Model 2 to Figure 4—figure supplement 1, where it is used to illustrate the generic aspects of competition between waves generated by spatially heterogeneous frequency profiles.

9) How are the kymograph lines determined in e.g. Figure 5? It is not clear if they are fit by hand, by computer, etc. and as a result, one is led to wonder how solid the conclusions from Figure 5 are. Additionally, the subtraction procedure in the blue-colored sub-panel of Figure 5A, B, C seems arbitrary.

Kymograph lines are determined by using a custom-made code that allows drawing lines by visually judging the disappearance of nuclei. The. tiff files are imported in Mathematica and for all *𝑥* the maximum intensity over the width is calculated. This allows us to have a one-dimension intensity profile for each time point. Kymographs as in Figure 1A and Figure 1—figure supplement 2were made from these profiles over time. Lines are drawn through the points of mitotic entry (disappearance of nuclei), for every visible cycle. The period is calculated by taking 20 points on these lines and determining the time to the next line. This gives an average period (and standard deviation) for each cycle. The locations of the nuclei (one-dimensional) are extracted from these kymographs at the last one or two lines (if nuclei are well-separated). For each nucleus the average distance to their neighbors (left and right) is calculated which is also plotted in Figure 1C and Figure 1—figure supplement 2. For the last two cycles, the maximum intensity over the cycle is calculated at every *x*, yielding an intensity profile at each cycle.

The procedure used in Figure 5 is elaborately discussed in the section Image Analysis, where we also show the robustness of the findings by changing the fitting parameters used in the analysis over a wide range.

10) There are several issues regarding the discussion of the boundary and/or dimensionality. First, the authors explore the effect of the boundaries at the ends of the tubes, but there is also a boundary azimuthally. Is there any way to determine whether this boundary plays a role in the initiation of the waves? Secondly, it is not clear what sets the length scale of the boundary-driven waves. Why is there a transition around 300 micron thick tubes? (This latter point may not be easy to answer from first principles.) Lastly (minor), around the fourth paragraph of the subsection “Wider systems lead to boundary-driven mitotic waves”, the authors discuss a two-dimensional system. They assert that the height of the droplet of extract relative to the diameter of the droplet determines the system dimensionality; this is not accurate; rather, the height of the droplet should be compared to the length scale of the wave (see a recent preprint for details: https://www.biorxiv.org/content/10.1101/2019.12.27.887273v1).

In our modeling in Figure 3D,F we found that the width of the transverse direction indeed plays an important role in determining the strength of a build-up of cell cycle regulators at the different boundaries. While for thinner tubes (especially 100, 200 μm width) there is not much build-up of regulators at the boundary in the transverse direction (but only in the longitudinal one), in thicker tubes that have several rows of nuclei such build-up becomes more pronounced. As soon as the two directions become of similar size, waves can emerge from the different boundaries (see Figure 3F and Video 2).

What determines the critical length scale of the tubes that lead to a transition to boundary-driven waves remains an open question. We believe that this is linked to the typical length scale of the internuclear distance and the nuclear domain (approx. 150 μm). As soon as tubes get wide enough (wider than approx. 2 x 150 = 300 μm), multiple rows of nuclei fit along the transverse direction. We believe that this increases the regularity of the nuclear pattern, which increases the likelihood of observing boundary-driven waves.

Regarding the comment about dimensionality: we thank the reviewer for pointing this out. We have rephrased our statements.

[Editors' note: further revisions were suggested prior to acceptance, as described below.]

All three reviewers agree that the paper is significantly improved by your new experiments, analyses and edits. However, there are few points that still need to be addressed before the paper can be accepted. One point (point 1) might require a bit more analysis or could be addressed by writing cautionary notes about the strength of conclusions that can be drawn from the uptake model. The remaining points (point 2-5) should all be addressed by textual modifications.1) There are still some logical gaps in the paper which impact the readability and theory-to-experiment correspondence. For instance, there is still not a strong quantitative link between the nuclear import model presented in Figure 3 and the resulting wave dynamics explored elsewhere. Panels D and E in Figure 3 show that uptake competition is the main driver of concentration increases at nuclei near the boundaries of a given system, at least within their uptake model. What remains unclear is how robust their conclusions are – i.e., whether their conclusions hold for other plausible uptake models. Thus, it remains unclear how universal their uptake model conclusions are – in other words, whether the conclusions hold for other plausible modeling choices. It would be useful to show that either (a) quantitative evidence that their modeling choices provide testable predictions that are validated in experiment or (b) many model classes lead to the qualitative phenomena they observe. If these analyses cannot be performed the authors should at least comment on the fact that their results hold for a specific choice of the uptake model and that it remains unclear how generalizable the results are.

We want to make a clear distinction between the models used for Figure 3 and Figure 4. In the first, the redistribution of cell cycle regulators due to a potential (mimicking nuclear import) is modeled, where the cell cycle period is fixed. This is done to show how regulators can build up at different locations depending on nuclear distribution and system dimensions, but it does not include any wave dynamics. It only shows the distribution of regulators if the cell cycle period would **not** be influenced by the local concentration of regulators. When we doinclude this feedback, we arrive at the model of Figure 4, which clearly shows waves. It also provided us with several predictions that were then qualitatively confirmed by our experiments. At the moment, we do not attempt a quantitative comparison as there are too many system parameters that remain unknown and experimentally uncharacterized.

It is indeed a relevant question whether our model is universal enough such that the qualitative conclusions hold for various plausible nuclear uptake models. The uptake model, without feedback to the cell cycle period (Figure 3), is already very general in the sense that import is modeled via a potential function, without the need of specifying how this potential is formed. The derivative of the potential is equal to the force, and together with diffusion they determine the full dynamics. There are no specific assumptions made regarding the experimental setup of cell-free extracts in the model itself, except for sizes and distances that are defined via model parameters. This means that all uptake models (in biology, physics, chemistry, …) that can be reduced to a potential function and diffusion will follow these dynamics within a certain range of parameters. We checked the influence of the parameters in the model in multiple (supplementary) figures, showing robustness within the given parameter ranges relevant to our experiments. Moreover, we have verified that multiple different shapes of the potential lead to similar results (not shown).

Finally, we also show in the supplementary figures of Figure 4 that the dynamics of competing pacemaker regions in reaction-diffusion systems are generic, always showing a similar transition between boundary waves and internal waves. Indeed, we showed that this behavior persisted in biochemically motivated models and universal models such as the FitzHugh-Nagumo system.

Altogether, this gives us confidence that the qualitative behavior (build-up of regulators at boundary due to nuclear uptake in systems with many nuclei + boundary waves in spatially coupled systems where those regulators increase the local oscillation frequency) is general. We now mention the generic character of our models in the main text.

2) Some of the analysis is still somewhat opaque. For instance, there is not a clearly visible wave in Figure 2C. It is also not clear how the choices to draw (or not draw) wave fronts are made in the analysis leading to Figure 5 – which is a central point of the paper; for instance, in panel C of Figure 5—figure supplement 1, the wave fronts could be drawn in multiple ways. In response to our concerns, the authors have noted their image analysis methods include "drawing lines by visually judging the disappearance of nuclei" and that they "show the robustness of the findings by changing the fitting parameters used in the analysis over a wide range". It is still not clear whether these lines are drawn manually or in a regimented, computerized fashion. We trust that they have done this procedure carefully enough that their conclusions would not be affected by whatever handling method they have chosen, but it would be nice to have a clearer description of what has been done as well as a more substantive supplementary dataset, e.g., an expanded version of Figure 5—figure supplement 1.

We agree that it is hard to see wave propagation in Figure 2C. However in the included Video 1 of the same experiment this is much clearer. We have stressed this in the text, such that a reader is not confused by Figure 2C and knows that the wave is most visible in Video 1.

The reviewer correctly points out that we can still explain in more detail the methodology that we use to calculate wave properties by drawing lines through a kymograph. We have now added panels D and E to old Figure 5—figure supplement 4 (new Figure 1—figure supplement 1), and have used those additional figures to better explain how we can get wave speeds from the microscopy data and we show that the potential error in wave speed estimates using this method is small.

Finally, we have added a new Figure 5—figure supplement 2 where we include additional kymographs of thicker (560um wide) tubes where we drew the lines using the method explained before. This figure helps to show that defining whether a wave is boundary-driven or internally-driven is unambiguous. All thicker tubes convincingly show waves that come from the boundary.

3) The new data on the dependency of mitotic wave speed on cell cycle timing shown in Figure 2A are very interesting. The observed dependency is difficult to reconcile with trigger waves and rather suggests a transition from sweep to trigger waves as the cell cycle slows down, as suggested by Vergassola et al., 2018. That must be stated. For example, authors could say: "The dependency of mitotic wave speed on cell cycle duration is consistent with a transition from sweep waves to trigger waves as cell cycle slows down, as proposed in Vergassola et al., 2018".

We have added the suggested comment.

*4*) The claims on the relevance of this paper findings to *Drosophila* must be further toned down. Even if the authors simulated sweep waves the internuclear distances used in the simulations are at least an order of magnitude higher than what is observe in *Drosophila*. So, in the fifth paragraph of the Discussion the authors should say something like: "However, the internuclear distance of our simulations is significantly larger than the one observed in the more crowded *Drosophila* embryo, so it remains unclear whether our results can directly extend to that system".

Indeed, we currently mainly use model parameters that are motivated by sizes and distances as observed in our experiments with *Xenopus* cell-free extracts. Although we have verified the generality of our results over a wide range of parameters, we have not changed the internuclear distance down to a scale of approx. 10 um, the approximate internuclear distance in *Drosophila* embryos. We have therefore added the suggested statement to stress that our results cannot directly be extended to *Drosophila* and this remains a question to be addressed in further research.

5) The importance of nuclear domains in *Drosophila* is an old concept named "energid". The authors should be clear about that in the second paragraph of the subsection “A computational model where nuclei spatially redistribute cell cycle regulators predicts the location of pacemaker regions”. It is also a bit misleading that the authors seem to imply that the nuclear domain of syncytial muscle or *Drosophila* embryos are similar in size to the ones observed in their experiments. They should comment on those importance size differences

We thank the reviewer for pointing out that there are important differences in size between the nuclear domain in *Drosophila* embryos and the nuclear domain in *Xenopus* cell-free extract. We now stress such differences in the manuscript.